# The Impact of Enforcing Representational Consistency of Identical Transformations for Disentangled Representation

## Abstract

Recent symmetry-based approaches in Variational Autoencoders (VAEs) have advanced disentanglement learning and compositional generalization. However, existing methods can encode identical semantic transformations differently depending on the specific sample pairs, which reduce the *representational consistency of identical transformations*. In this paper, we analyze how three commonly used symmetry parameterization families in prior work, namely (1) matrix-exponential parameterizations over the general linear group $GL(n)$, (2) vector-additive actions in latent space, and (3) surjective mappings from latent vectors to the unit circle, can make it difficult to represent identical transformations consistently in dimension-wise disentangled latent spaces. To address this issue, we propose a framework that maps latent vectors to a bijective cyclic representation on the unit circle via the Cayley transform, together with a fixed-grid codebook regularization. We study this problem in a controlled setting and develop practical weakly supervised and supervised variants. Experiments on disentanglement benchmarks and compositional generalization tasks show that the proposed framework yields improved disentanglement performance and strong compositional generalization under supervised settings, with the stronger-supervision variants providing empirical reference points for the representational capacity of the framework. Overall, our results suggest that consistent representation of identical transformations is a useful design principle for improving disentanglement and generalization performance in the considered setting.

## 1 Introduction

Disentangled representation learning (Bengio et al., 2013; Wang et al., 2023) is a central problem in representation learning, as factorized latent representations can improve generalization under novel contexts and support more reliable reasoning over the underlying factors of variation (Montero et al., 2021). Since Locatello et al. (2019) establish that unsupervised disentanglement learning is impossible without suitable inductive bias (Bengio et al., 2013; Higgins et al., 2018), a major line of research focuses on introducing inductive biases that encourage latent variables to align with the factors of variation.

Among such approaches, symmetry-based modeling emerge as a promising direction. In particular, several recent works incorporate equivariant function modeling into Variational AutoEncoders (VAEs) by explicitly defining symmetry groups and their actions in latent space (Higgins et al., 2018). Concretely, prior methods instantiate specific symmetry parameterizations, including matrix Lie groups with matrix multiplication (Zhu et al., 2021; Keurti et al., 2023; Jung et al., 2024; Winter et al., 2022), cyclic-group formulations induced by surjective projections to the unit circle (Yang et al., 2021; Tonnaer et al., 2022; Cha & Thiyagalingam, 2023), and vector-additive constructions (Balabin et al., 2024). These studies show that symmetry-aware equivariant modeling improves disentanglement without supervision, thereby highlighting the usefulness of symmetry representations as an inductive bias.

However, an important issue remains insufficiently addressed: even when the underlying transformation is identical, existing approaches encode that transformation differently depending on which pair of samples induces it (Hwang et al., 2023). Such pair-dependent symmetry representations are undesirable for disentanglement

learning, because disentangled latent spaces are expected to capture the same factor change in a stable manner across different instances (Higgins et al., 2018). More broadly, this behavior contrasts with how humans tend to conceptualize transformations: the same transformation is often understood as a content-independent rule, or relational schema, that can be applied consistently across different contexts (Marcus et al., 1999; Gentner, 1983). Motivated by this perspective, we argue that an effective symmetry representation for disentanglement should encode the identical transformations consistently, independently of the particular sample pair from which it is inferred. **(S 3-1 & MA 1)** Importantly, this requirement is not merely intuitive: in Section 3.2, we provide a controlled empirical example showing that a lack of representational consistency leads to a clear failure in learning disentangled representations.

This consistency issue naturally raises a more fundamental question: what form of symmetry representation is appropriate for learning disentangled representations? Although prior works show the usefulness of symmetry-based inductive bias, they offer limited analysis of which symmetry parameterization is most suitable for consistently encoding identical transformations. Moreover, existing approaches often adopt a specific representation without thoroughly comparing how alternative parameterizations influence the consistency and suitability of the latent symmetry representation. We argue that this missing analysis is important because the choice of parameterization can directly influence whether identical transformations are represented consistently in latent space. Since our goal is to isolate and evaluate the role of symmetry parameterization itself, we study this question in a supervised setting, which provides a more controlled testbed than a fully unsupervised formulation and serves as a necessary step toward future unsupervised extensions.

In this work, we address these issues jointly: (1) the lack of mechanisms that enforce representational consistency of identical transformations, and (2) the limited analysis of which symmetry parameterizations are appropriate for achieving such consistency in disentanglement learning. To this end, we first analyze three commonly used symmetry parameterization families in prior work: matrix-exponential parameterizations over the general linear group (Zhu et al., 2021; Keurti et al., 2023; Jung et al., 2024; 2026; Winter et al., 2022), vector-additive actions in latent space (Balabin et al., 2024; Hwang et al., 2023), and surjective mappings from latent vectors to the unit circle (Yang et al., 2021; Tonnaer et al., 2022; Cha & Thiyagalingam, 2023). We show that these formulations have intrinsic limitations in representing identical transformations consistently in disentangled latent spaces. Building on this analysis, we propose a bijective representation on the unit circle based on the Cayley transform (Kreyszig et al., 2011), which provides a more suitable basis for consistent symmetry encoding. In practice, we further realize this principle through a fixed grid (codebook) (Hsu et al., 2023), which guides the model to represent identical transformations in latent space.

Our main contributions are as follows:

1. We identify a previously underexplored issue in symmetry-based disentanglement learning, the representational inconsistency of identical transformations across different sample pairs, and theoretically show that this issue is closely tied to the choice of symmetry parameterization by analyzing three widely used parameterization families from prior work.

2. We propose a bijective symmetry representation on the unit circle, together with a codebook-based latent vector space, to guide the model toward representational consistency of identical transformations.

3. We develop a practical learning framework that induces equivariance while promoting symmetry representation consistency, and empirically evaluate its effects on disentanglement learning and compositional generalization controlled benchmarks.

## 2 Preliminaries: Group Theory

**Binary operation:** Binary operation on a set $S$ is a function that $* : S \times S \to S$, where $\times$ is a cartesian product.

**Group:** A group is a set $G$ together with binary operation $*$, that combines any two elements $g_a$ and $g_b$ in $G$, such that the following properties:

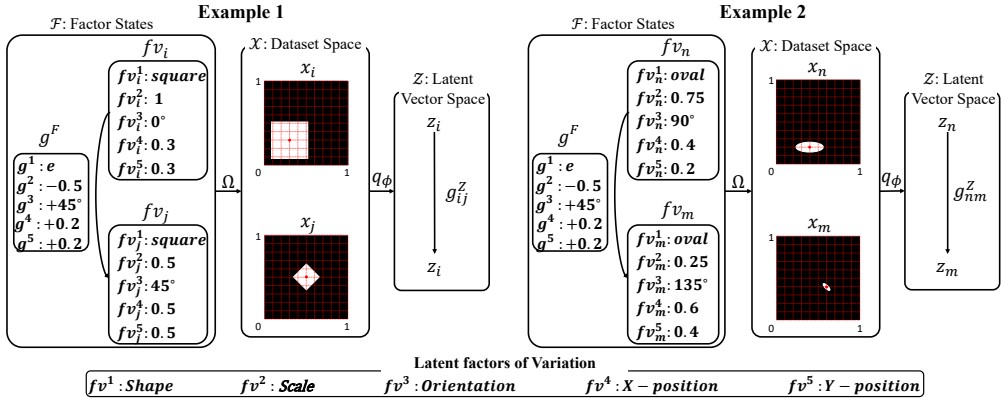

Figure 1: Representational consistency of identical transformations between example 1 and example 2 in mapping factor states, datasets, and latent representations. Equivariant function: $h_\phi := q_\phi \circ \Omega$ satisfies $g_i^z \cdot h_\phi(fv_i) = h_\phi(g_i^F \cdot fv_i)$. Representational consistency of identical transformations: the same symmetry $g^F$ is acted on latent factors $fv_i, fv_n$, and the symmetry acted on the latent vector space is represented as a single symmetry $g_{ij}^Z = g_{nm}^Z = \Gamma(g^F)$ regardless of pairs.

- closure: $g_a, g_b \in G \Rightarrow g_a * g_b \in G$.

- Associativity: $\forall g_a, g_b, g_c \in G,\ s.t.\ (g_a * g_b) * g_c = g_a * (g_b * g_c)$.

- Identity element: There exists an element $e \in G,\ s.t.\ \forall g \in G, e * g = g * e = g$.

- Inverse element: $\forall g \in G, \exists g^{-1} \in G:\ g * g^{-1} = g^{-1} * g = e$.

**Group action:** Let $(G, *)$ be a group and set $X$, binary operation $\cdot : G \times X \to X$, such that following properties:

- Identity: $e \cdot x = x$, where $e \in G, x \in X$.

- Compatibility: $\forall g_a, g_b \in G, x \in X, (g_a * g_b) \cdot x = g_a \cdot (g_b \cdot x)$.

**Group Homomorphism:** Let $(G, *))$, and $(H, \cdot)$ be two groups. A function $\Gamma : G \to H$ is a group homomorphism if for all elements $g_a, g_b \in G$, it satisfies the following equation: $\Gamma(g_a * g_b) = \Gamma(g_a) \cdot \Gamma(g_b)$.

**Equivariant map.** Let $G$ act on $\mathcal{X}$, let $H$ act on $\mathcal{Y}$, and let $\Gamma : G \to H$ be a group homomorphism. A map $f : \mathcal{X} \to \mathcal{Y}$ is $\Gamma$-equivariant if $f(g \cdot x) = \Gamma(g) \cdot f(x), \forall g \in G, x \in \mathcal{X}$.

**Group Isomorphism:** Given two groups $(G, *)$ and $(H, \cdot)$, a group isomorphism is a bijective function $f : G \to H$ that satisfies as follows: $f(u * v) = f(u) \cdot f(v)$, where $\forall u, v \in G$ $((G, *) \cong (H, \cdot))$. The factor group $\mathbb{R}/\mathbb{Z}$ and the circle group $S^1$ are isomorphic, where group $\mathbb{Z}$ of integers with addition, and $S^1 = \{z \in \mathbb{C} : |z| = 1\}$. $(\mathbb{R}/\mathbb{Z} \cong S^1)$. More details are in Appendix A.

## 3 Representational Consistency of Identical Transformations

### 3.1 Equivariant Map from Factor States to Latent Vector Space

**Factor States.** The factor (world) states is referred to from Higgins et al. (2018) to define the equivariant function between the factor states and latent vector space for disentanglement learning with symmetries. We then follow the factor stats as :

- Factor States $\mathcal{F}$ composed with latent factors of variation $fv = fv^1 \times fv^2 \times \cdots \times fv^k$, and $F = F^1 \times F^2 \times \cdots \times F^k$, where $fv^i \in F^i$, $fv \in F$, and set of factors $F \subset \mathcal{F}$ as shown in Higgins et al. (2018) and Figure 1.

**Symmetries in Factor States.** $G^F = G^{F^1} \times G^{F^2} \times \cdots \times G^{F^k}$ is acted on the factor states $\mathcal{F}$, *e.g.*) $fv_j = g^F \circ fv_i$ as shown in Figure 1, where $g^F \in G^F$, and $fv_i, fv_j \in F$. Function $\Omega$ maps factor states to dataset space $\Omega : \mathcal{F} \to \mathcal{X}$ and function $q_\phi$ maps dataset to latent vector space $q_\phi : \mathcal{X} \to \mathcal{Z}$ as shown in Figure 1. Let the composite function $h_\phi := q_\phi \circ \Omega : \mathcal{F} \to \mathcal{Z}$.

**Equivariant Function from Factor States to Latent Vector Space.** Modeling equivariant functions is a common approach for disentanglement learning (Zhu et al., 2021; Yang et al., 2021; Keurti et al., 2023; Jung et al., 2024) and compositional generalization (Hwang et al., 2023) to inject inductive bias. As follows the definition of equivariant function $f$ in Section 2, $h_\phi$ satisfies $g^Z \cdot h_\phi(fv) = h_\phi(g^F \cdot fv)$, where $G^F, G^Z$ are symmetry group acted on space $\mathcal{F}, \mathcal{Z}$ respectively, and $g^F \in G^F, g^Z \in G^Z$.

### 3.2 (MI 1) Representational Consistency of Identical Transformations

Let $G^F$ be a group acting on the factor space $\mathcal{F}$, and let $G^Z$ be a group acting on the latent space $\mathcal{Z}$.

**Definition 3.1. Representational consistency of identical transformations.** Let $g^F \in G^F$ be a factor space transformation, and consider two pairs $(fv_i, fv_j), (fv_n, fv_m) \in \mathcal{F} \times \mathcal{F}$ generated by the same transformation $g^F$: $fv_j = g^F \cdot fv_i$ and $fv_m = g^F \cdot fv_n$. Let $z_i, z_j, z_n, z_m \in \mathcal{Z}$ be their latent representations: $z_i = h'_\phi(fv_i), z_j = h'_\phi(fv_j), z_n = h'_\phi(fv_n)$, and $z_m = h'_\phi(fv_m)$, where $h'_\phi : \mathcal{F} \to \mathcal{Z}$. Let $g^Z_{ij}, g^Z_{nm} \in G^Z$ denote the pair-induced latent transformations satisfying $z_j = g^Z_{ij} \cdot z_i$ and $z_m = g^Z_{nm} \cdot z_n$. We say that the same factor space transformation $g^F$ is represented consistently across two pairs if the corresponding pair-induced latent transformations are identical:

$$g^Z_{ij} = g^Z_{nm}. \tag{1}$$

We refer to this property as the *representational consistency of identical transformations* (RCIT).

**Limited Guarantee of Consistency in Disentangled Space.** Previous works (Zhu et al., 2021; Yang et al., 2021; Jung et al., 2024; Hwang et al., 2023) formulate equivariant mappings between the dataset space and the latent vector space. Although Hwang et al. (2023) shows improved coherence for the same transformation, the problem of consistently representing identical transformations still remains unresolved. This suggests that equivariance alone does not necessarily guarantee the consistent representation of identical transformations defined above. Therefore, we explicitly treat consistency as an additional requirement for learning disentangled representation, and discuss the limitations of previous formulations in Section 4.

(**S 3-1, MA 1** & **W 1, MA 1, MI 1**) **Motivation: Impact of RCIT.** To isolate the effect of consistency supervision, we compare $S^2/S$-$\beta$-VAE Locatello et al. (2020b) with CTFG-SP (w/o FG) under a controlled setting. Both methods use the same encoder, decoder, training data, factor labels, and amount of supervision. Neither method employs the fixed-grid component or any additional architectural inductive bias. Transformation steps of the CTFG-SP are computed solely from differences between the same factor labels used by $S^2/S$-$\beta$-VAE.

Table 1: Disentanglement results on dSprites.

| Method | Inductive Bias | | | dSprites (DL) | | | |
|---|---|---|---|---|---|---|---|
| | Supervision | Architecture | RCIT | $\beta$-VAE | FVM | MIG | DCI |
| $S^2/S$-$\beta$-VAE | Factor label | ✗ | ✗ | 75.71(±11.16) | 70.34(±14.01) | 7.59(±5.80) | 11.52(±6.14) |
| CTFG-SP (w/o FG) | Transformation | ✗ | ✓ | **90.75**(±5.23) | **92.36**(±2.54) | **49.06**(±5.46) | **62.55**(±6.55) |

As shown in Table 1, the comparison between $S^2/S$-$\beta$-VAE and CTFG-SP (w/o FG) provides empirical motivation for the representational consistency of identical transformations. The $S^2/S$-$\beta$-VAE baseline is trained using factor-level consistency (ground-truth factor labels), whereas CTFG-SP (w/o FG) uses transformation-level consistency supervision without the fixed-grid component. Neither model employs an additional architectural inductive bias. CTFG-SP (w/o FG) substantially outperforms $S^2/S$-$\beta$-VAE across all disentanglement metrics, suggesting that the consistent representation of identical transformations provides an effective inductive bias for disentanglement learning.

Figure 2 qualitatively supports this observation through latent traversals. In each row, we vary a single latent dimension while holding all remaining dimensions fixed. Under factor-label supervision, a single latent traversal simultaneously changes shape, scale, and y-position, indicating that these factors remain entangled. In contrast, transformation-level consistency supervision produces factor-specific trajectories: varying the

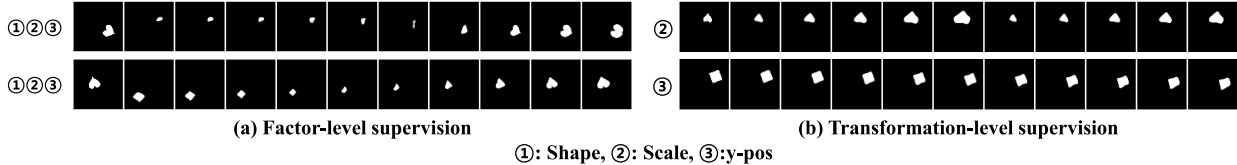

(a) Factor-level supervision        (b) Transformation-level supervision

①: Shape, ②: Scale, ③:y-pos

Figure 2: Latent traversals on dSprites under (a) factor-level supervision and (b) transformation-level supervision. The circled numbers denote the varying factors: ① shape, ② scale, and ③ y-position.

latent dimension associated with scale or y-position predominantly changes only the corresponding factor while preserving the others. These results illustrate how the lack of representational consistency can lead to entangled latent trajectories.

# 4 Limitations of Symmetry Parameterizations for RCIT in Dimension-Wise Disentangled Representations of Finite Cyclic Factors

**Cyclic Structure of Latent Factors of Variation.** To make Definition 3.1 operational, we must specify a concrete symmetry group acting on the factor states. In standard disentanglement benchmarks (dSprites, 3D Shapes, MPI3D), the labels of latent factors of variation are represented as discrete integers in a finite range, e.g., $[0, k-1]$, and the same interval difference between labels corresponds to the same semantic change in the dataset space (e.g., a fixed rotation angle, a fixed position shift). When such a factor admits a modulo structure, its label space is naturally modeled by the cyclic group $\mathbb{Z}_k$. We therefore assume that such latent factors of variation are isomorphic to cyclic groups, and formulate the symmetry group acting on factor states as $G^F = G^{F^1} \times G^{F^2} \times \cdots \times G^{F^k}$, $G^{F^i} \cong \mathbb{Z}_{|F^i|}$, where $|F^i| \in \mathbb{Z}^+$ is the number of distinct values of the $i$-th factor. This assumption holds for all datasets used in this paper.

**Lack of Analysis on Suitable Symmetry Parameterizations for Disentangled Representations.** The goal of dimension-wise disentangled representations (Bengio et al., 2013; Wang et al., 2023) with symmetries ($g_z \in G_z$) is to allow each factor symmetry to change a single latent dimension value. ==As described above, the symmetries of the factor states are modeled as a product of finite cyclic groups. However, there is limited analysis of which latent symmetry parameterizations can represent these cyclic transformations while satisfying the representational consistency of identical transformations across different sample pairs in the disentangled space.== In this section, we analyze three commonly used symmetry parameterization families under the stated finite-cyclic and dimension-wise assumptions, with complete proofs provided in Appendix B. Specifically, cases 1 and 2 admit only the trivial identity action under their respective parameterization constraints, while case 3 can map distinct factor symmetries to the same latent symmetry and thereby lose part of the factor information.

## 4.1 (**W 1** & **MA 1** & **MI 2**) Scope and Conditions of the Analysis

Before presenting the three cases, we explicitly state the conditions defining the scope of our analysis.

**Condition 4.1.** (Finite Cyclic Factor Symmetries) We adopt the finite cyclic factor symmetry defined in Section 4.

**Condition 4.2.** (Dimeionsion-Wise Disentangled Representation) Each non-identity factor symmetry acts only on its corresponding latent dimension. Specifically, for every $g^Z \in G^Z \setminus \{e\}$ associated with the $i$-th factor and every $z \in Z$, $(g^Z \cdot z)^j = z^j$ for all $j \neq i$, $(g^Z \cdot z)^i \neq z^i$.

**Condition 4.3.** (Global and Structure-Preserving Latent Action) The mapping $\Gamma : G^F \to G^Z$ is an injective group homomorphism that preserves the composition and cyclic structure of the factor symmetries. Moreover, the latent representation satisfies RCIT as defined in Definition 3.1.

## 4.2 Formal Analysis of Symmetry Parameterizations

**Case 1: A Limitation of Matrix-Exponential Parameterizations over $GL(n)$.** Matrix-exponential parameterizations over the general linear group $GL(n)$, as used in prior works (Zhu et al., 2021; Kuzina et al., 2022; Miyato et al., 2022; Marchetti et al., 2023; Jung et al., 2026), are limited in achieving a representational consistency of identical transformations for disentangled representations. We first characterize the behavior of dimension-wise disentangled representations under matrix-exponential parameterizations, and then show the resulting limitation of $GL(n)$.

**Proposition 4.4.** *Let the symmetry group $G_z$ $(GL'(n))$ is defined as a subgroup of the General Linear group that implemented with matrix exponential, where $GL'(n) = \{e^{\boldsymbol{M}} | \boldsymbol{M} \in \mathbb{R}^{n \times n}\}$, $g^k$ is an element of $GL'(n)$, and $g = \prod_k g^k$. Then $e^g \boldsymbol{z} = e \boldsymbol{I} g \boldsymbol{z} + \boldsymbol{v}'$.*

**Theorem 4.5.** *(Limitation of $GL'(n)$) By Proposition 4.4, if $g \in GL'(n)$ is compatible with Conditions 4.1–4.3 and, $|G^{F^i}| > 2$, then $g = \boldsymbol{I}$.*

**Theorem 4.6.** *(Limitation of $GL(n)$) If Conditions 4.1–4.3 hold, $|G^{F^i}| > 2$, and $H = \{h \in GL(n) \mid h = \boldsymbol{I} + \boldsymbol{M}^i\}$, then $h = \boldsymbol{I}$ is the only element of $H$.*

Therefore, the limitation of $GL(n)$ is that only the $\boldsymbol{I}$ represents the identical transformations with disentangled representation according to the Theorem 4.5, and Theorem 4.6. It implies that if $g \neq \boldsymbol{I}$, then $g$ can not represent the identical transformations. More details are in Appendix B.2.

**Case 2: A Limitation of Vector-Additive Actions.** Another commonly used family is vector-additive actions in latent space, where the group action between two latent vectors is defined by vector addition, as in Balabin et al. (2024). We show that this family also has limitations in maintaining a representational consistency of identical transformations for cyclic groups.

**Corollary 4.7.** *If the group action is defined as $act(g, \boldsymbol{z}_i) = g + \boldsymbol{z}_i$, then only zero vector represent the identical transformations for cyclic group with disentangled representation, where $\boldsymbol{z} \in \mathbb{R}^n$.*

According to the Corollary 4.7, it also shows a limitation in that only the identity element $\vec{0}$ represents the identical transformations. More details of the proof are in Appendix B.3.

**Case 3: A Limitation of Surjective Mappings to the Unit Circle.** The third family consists of surjective mappings from latent vectors to the unit circle (Yang et al., 2021; Tonnaer et al., 2022; Cha & Thiyagalingam, 2023). We show that this family can cause undifferentiated symmetries under more general latent factors of variation and can lose part of the dataset's factor information.

**Proposition 4.8.** *(**W 2** & **MA 2**) Let $b : Z \to \mathcal{Y}$ be surjective but non-injective. Suppose that there exist two factor states $fv_i$ and $fv_j$ that differ in the $k$-th factor and satisfy $b(h_\phi(fv_i)) = b(h_\phi(fv_j))$. Then $b \circ h_\phi$ cannot distinguish these two values of the $k$-th factor and therefore does not provide a faithful dimension-wise representation of that factor.*

(**W 2** & **MA 2**) Therefore, a surjective but non-injective mapping can lose factor-relevant information when it maps factor states with different values of the same factor to an identical representation. In this case, the resulting representation cannot distinguish all values of that factor and fail to provide a dimension-wise disentangled representation. This result concerns factor-state distinguishability. Further details are provided in Appendix B.4.

# 5 Method

## 5.1 Motivation: From Limitations to Bijective Cyclic Representation

The above analysis reveals two distinct types of limitation. Cases 1 and 2 exhibit a *Type I* failure: the consistent representation condition in Definition 3.1 is satisfied only by the trivial identity element, making non-trivial consistent encoding impossible. (**W2** & **MA 2**) Case 3 exhibits a *Type II* failure: a surjective but non-injective mapping can map factor states with different values of the same factor to an identical

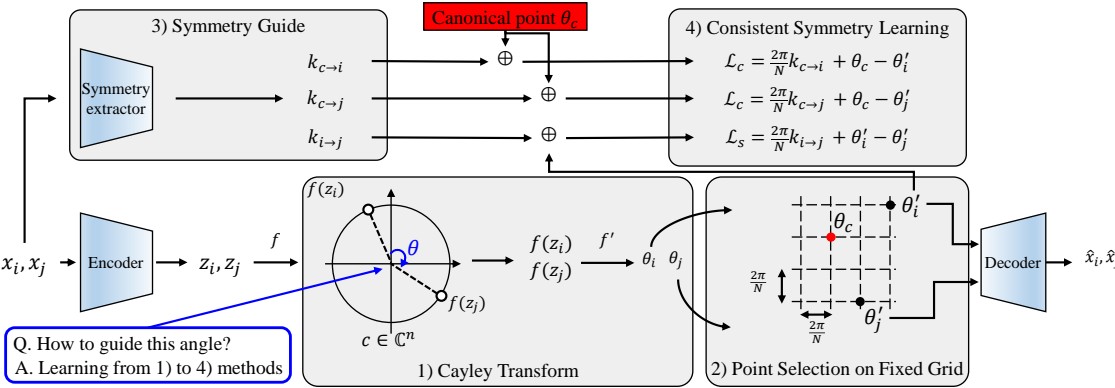

Figure 3: The overall architecture of our proposed method comprises four distinct components: 1) Cayley transform of latent vectors to angle space, 2) point selection of fixed grid for consistent symmetry, 3) defining the step size between two inputs through three methods, and 4) a loss function that satisfies the group action $act(g, \theta) = g + \theta$.

representation. This collision loses factor-relevant information and prevents the resulting representation from distinguishing all values of that factor. Together, these results motivate a representation that is (i) bijective to preserve the full symmetry structure, and (ii) discretely structured so that identical transformations map to the same grid point regardless of the sample pair. We address both requirements in the following subsections, and formally show in Appendix C.1 that the resulting framework satisfies Definition 3.1 under the cyclic group structure.

## 5.2 Cayley Transform: Representational Consistency of Identical Transformations on Cyclic Representations

We first define the latent vector space as a $G$-set of a cyclic group to ensure the representational consistency of identical transformations because 1) we assume that $G^{F^i}$ is a cyclic group as referred in Section 4 and 2) a single cyclic group element can represent all elements, as demonstrated in Higgins et al. (2018). To address the issues discussed in Section 4, we implement the Cayley transform (Kreyszig et al., 2011) that maps real numbers to complex numbers bijectively.

**Cyclic Group for Consistency.** As we assume that factor states serve as latent factors of variation for input samples in Section 4, the symmetry group $G^F = \mathbb{Z}/|F^1|\mathbb{Z} \times \mathbb{Z}/|F^2|\mathbb{Z} \times \ldots \times \mathbb{Z}/|F^k|\mathbb{Z}$, where $|F^i| \in \mathbb{Z}^+$ represents the number of factors in the datasets. Additionally, the cyclic group effectively represents the symmetries over the same group action, as the cyclic group $G = \{e, g^1, g^2, \ldots, g^{n-1}\}$ consists entirely of integer powers of the group element $g$. Therefore, if the model learns a single symmetry element $g$, it represents the entire symmetry group. Motivated by Yang et al. (2021), we implement the cyclic group as the $n^{th}$ root of unity.

**Group Action and $G^c$-set.** As we define the cyclic group as $n^{th}$ root of unity, the cyclic group $G^c = G^c_1 \times G^c_2 \times \cdots \times G^c_k$ and $G^c_i = \{g^c_i | g^c_i = \frac{2\pi}{N}k', k' \in \{0, 1, 2, \ldots, N-1\}\}$, where $N \in \mathbb{Z}^+$. We define the group action $act : \mathbf{\Theta}^n \times G^c \to \mathbf{\Theta}^n$ as $act(g^c, \boldsymbol{\theta}) = g^c + \boldsymbol{\theta}$, where the latent vector $\boldsymbol{\theta} \in \mathbf{\Theta}^n$, and $-\pi < \boldsymbol{\theta}^i \leq \pi$, where $\theta^k$ is a $i$-th dimension value of vector $\boldsymbol{\theta}$.

**Cayley Transform for Complex Number Space.** VAE frameworks establish the latent vector space in the real number space with the Gaussian normal distribution, so the latent vector space is not a $G^c$-set as we assume ($\boldsymbol{z} \in \mathbf{\Theta}^n$). For the defined $G^c$-set, we utilize the Cayley transform and invertible function $f' \circ f : [-\infty, \infty] \to \{\theta| - \pi < \theta \leq \pi\}$. To map real numbers to the complex number space, we utilize the Cayley transform function (bijective) $f : [-\infty, \infty] \to \{\boldsymbol{c}^i \in \mathbb{C} : |\boldsymbol{c}^i| = 1\}$, defined as follows:

$$f(\boldsymbol{z}^i) = \boldsymbol{c}^i = \frac{\boldsymbol{z}^i - i}{\boldsymbol{z}^i + i} = i\frac{-2\boldsymbol{z}^i}{(\boldsymbol{z}^i)^2 + 1} + \frac{(\boldsymbol{z}^i)^2 - 1}{(\boldsymbol{z}^i)^2 + 1}, \tag{2}$$

Table 2: Comparison of symmetry representations with CTFG-GT models. We evaluate ground truth supervised models under different symmetry representations to investigate which representation is better suited to consistently encoding identical transformations. Specifically, we compare three symmetry modeling approaches—General Linear group (GL($n$)), vector addition (Add.), and surjective mappings (Sur.)—against our proposed method.

| Symmetry | 3D Shapes | | | | dSprites | | | |
|---|---|---|---|---|---|---|---|---|
| | beta-VAE | FVM | MIG | DCI | beta-VAE | FVM | MIG | DCI |
| CTFG-GT ($GL(n)$) | 73.50($\pm$17.92) | 66.16($\pm$15.95) | 19.66($\pm$24.29) | 37.93($\pm$23.92) | 65.11($\pm$3.89) | 47.69($\pm$8.04) | 2.12($\pm$1.73) | 5.90($\pm$2.11) |
| CTFG-GT (Add.) | 78.60($\pm$11.43) | 60.50($\pm$15.29) | 25.13($\pm$17.12) | 45.84($\pm$8.36) | 68.89($\pm$11.45) | 68.31($\pm$9.48) | 7.22($\pm$4.31) | 11.83($\pm$3.85) |
| CTFG-GT (Sur.) | 73.27($\pm$9.97) | 63.85($\pm$6.81) | 6.20($\pm$4.03) | 29.79($\pm$9.86) | 80.60($\pm$10.83) | 60.40($\pm$6.47) | 13.68($\pm$3.86) | 23.01($\pm$2.28) |
| CTFG-GT | **100.00**($\pm$0.00) | **100.00**($\pm$0.00) | **96.57**($\pm$0.80) | **99.94**($\pm$0.18) | **95.80**($\pm$4.57) | **99.26**($\pm$1.12) | **51.81**($\pm$2.97) | **63.26**($\pm$2.73) |

where the $\boldsymbol{z}^i$ is a $i^{th}$ dimension value of $\boldsymbol{z} \in \mathbb{R}^n$. We define a bijective function that maps complex numbers to the angle space for simplicity $f' : \{\boldsymbol{c}^i \in \mathbb{C} : |\boldsymbol{c}^i| = 1\} \rightarrow \{\boldsymbol{\theta}^i| - \pi < \boldsymbol{\theta}^i \le \pi\}$ as follows:

$$\boldsymbol{\theta}^i = f'(\boldsymbol{c}^i) = \begin{cases} \cos^{-1}(\mathfrak{R}(\boldsymbol{c}^i)) - \pi, \text{ if } \mathfrak{I}(\boldsymbol{c}^i) >= 0 \\ \pi - \cos^{-1}(\mathfrak{R}(\boldsymbol{c}^i)) \text{ otherwise} \end{cases}, \tag{3}$$

where $\mathfrak{R}(\boldsymbol{c}^i)$ and $\mathfrak{I}(\boldsymbol{c}^i)$ are real and imaginary parts of $\boldsymbol{c}^i$, respectively.

### 5.3 Point Selection on Fixed Grid for Representational Consistency of Identical Transformations

To address the cyclic group and consistency of identical transformations in the latent vector space, we set the space as a fixed grid (Mentzer et al., 2024) instead of a continuous or learnable grid (Hsu et al., 2023). Because, as shown in Figure 3, the interval between two nearest codes is always $\frac{2\pi}{N_i}$, and it implies that equation $g_i^c = \frac{2\pi}{N_i}$ is satisfied for all cases. Then we utilized the finite scalar quantization (Mentzer et al., 2024) for fixed codebook $\boldsymbol{V} \in \mathbb{R}^N$ as follows:

$$\boldsymbol{V} = [-\pi + \frac{2\pi}{N}, \cdots, -\pi + \frac{2\pi}{N}k', \cdots, -\pi + \frac{2\pi}{N}(N-1), \pi]. \tag{4}$$

Then we select the nearest neighbor of the latent vector as Hsu et al. (2023): $\boldsymbol{\theta}'^j = \arg\min_{\boldsymbol{V}^i \in \boldsymbol{V}} |\boldsymbol{V}^i - \boldsymbol{\theta}^j|$, where $\boldsymbol{V}^i$ is the $i^{th}$ dimension value of the codebook $\boldsymbol{V}$. We define the grid loss $\mathcal{L}_{grid} = ||\boldsymbol{\theta}' - \boldsymbol{\theta}||_2^2$ to consistently select the gird, where $|| \cdot ||_2$ is a L2 norm.

### 5.4 Symmetry Guide: How to Select Step ($k'$)?

As we define the cyclic group $G_i^c = \{g_i^c | g_i^c = \frac{2\pi}{N_i}k', k' \in \{0, 1, 2, \ldots, N_i - 1\}\}$, we implement the step $k$ to guide how much step moves to be $\boldsymbol{\theta}_i = g^c + \boldsymbol{\theta}_j$ (defined group action). We propose three guiding approaches: 1) ground truth, 2) supervised, 3) and weakly-supervised methods.

**Ground Truth Based Method.** As shown in Figure 3, we set the symmetry group elements $g^c$ from the ground truth of samples as follows:

$$\boldsymbol{k}'_{i \rightarrow j} = \begin{cases} l_j - l_i & \text{if } l_j - l_i \ge 0 \\ N + l_j - l_i & \text{otherwise} \end{cases}, \tag{5}$$

where $\boldsymbol{k}'_{i \rightarrow j}$ is a step size, $g^c = \frac{2\pi}{N}\boldsymbol{k}'_{i \rightarrow j}$, and $l_i$ and $l_j$ are labels of samples $x_i$ and $x_j$, respectively.

**Supervised Method.** As shown in Figure 3, we train the symmetry extractor to predict the labels of samples ($\hat{l}$) using cross-entropy loss, defined as $\mathcal{L}_{pred} = C.E.(\hat{l}, l)$. We then define the symmetry group elements by $\hat{l}$ instead of the ground truth labels $l$.

**Weakly-Supervised Method.** We utilize a $p$ ratio of the labels for prediction, while the remaining labels are predicted using the pseudo-label loss as follows: $\mathcal{L}_{pl} = \sum_i^{|F_i|} D_{\mathrm{KL}}(p(l^i|x)||p(l^i))$, where the $l^i$ is a $i^{th}$ factor of the label and a discrete uniform distribution $p(l^i) \sim \mathcal{U}\{1, |F_i|\}$. We define the $p(l^i|x)$ as the distribution of the classifier.

Table 3: Disentanglement performance on dSprites, 3D Shapes, and the MPI3D dataset. (Unsup: unsupervised method, Weak-sup: weakly supervised, Sup: supervised, GT: Ground-Truth, CTFG: our method). Bold text indicates a higher value than the other baseline models.

| type | Method | dSprites | | | | 3D Shapes | | | | MPI3D | | | |
|---|---|---|---|---|---|---|---|---|---|---|---|---|---|
| | | beta-VAE | FVM | MIG | DCI | beta-VAE | FVM | MIG | DCI | beta-VAE | FVM | MIG | DCI |
| Unsup | $\beta$-VAE | 78.40($\pm$9.03) | 64.84($\pm$11.40) | 14.52($\pm$9.33) | 22.37($\pm$11.80) | 90.33($\pm$7.42) | 72.63($\pm$19.55) | 40.49($\pm$23.31) | 54.32($\pm$16.45) | 57.60($\pm$7.93) | 40.86($\pm$3.92) | 4.91($\pm$1.43) | 22.29($\pm$1.42) |
| | $\beta$-TCVAE | 81.80($\pm$11.91) | 70.16($\pm$12.41) | 19.03($\pm$9.40) | 30.89($\pm$8.96) | 88.20($\pm$7.91) | 74.45($\pm$14.68) | 43.17($\pm$28.28) | 59.71($\pm$14.79) | 55.40($\pm$9.52) | 40.80($\pm$2.60) | 5.23($\pm$1.96) | 21.47($\pm$2.35) |
| | Factor-VAE | 87.20($\pm$7.50) | 76.80($\pm$7.50) | 24.98($\pm$12.03) | 33.38($\pm$12.27) | 95.60($\pm$6.99) | 80.53($\pm$11.41) | 54.54($\pm$25.10) | 66.55($\pm$9.52) | 54.00($\pm$7.18) | 39.64($\pm$3.81) | 4.34($\pm$0.69) | 21.24($\pm$2.04) |
| | CLG-VAE | 88.40($\pm$5.80) | 82.21($\pm$5.73) | 20.89($\pm$7.40) | 29.96($\pm$7.05) | 86.20($\pm$5.61) | 77.36($\pm$7.99) | 50.39($\pm$12.37) | 59.25($\pm$11.21) | 46.40($\pm$7.35) | 37.31($\pm$2.27) | 20.77($\pm$5.70) | 24.26($\pm$2.73) |
| Weak-sup | Ada-GVAE | 83.60($\pm$2.61) | 83.67($\pm$2.97) | 21.34($\pm$5.35) | 47.26($\pm$1.89) | 72.75($\pm$6.50) | 59.81($\pm$6.14) | 24.77($\pm$7.48) | 64.57($\pm$4.04) | 64.89($\pm$7.22) | 46.10($\pm$3.19) | 22.48($\pm$8.14) | 41.30($\pm$7.00) |
| | CTFG-wSP | **90.50**($\pm$7.55) | **84.50**($\pm$1.41) | **31.95**($\pm$2.40) | 39.36($\pm$1.49) | **95.00**($\pm$7.07) | **88.81**($\pm$13.17) | **57.94**($\pm$16.52) | **72.14**($\pm$3.23) | **67.50**($\pm$12.68) | **81.59**($\pm$3.80) | **61.07**($\pm$4.81) | **78.15**($\pm$0.57) |
| Sup | CTFG-SP | **91.40**($\pm$4.99) | **93.74**($\pm$1.82) | **51.02**($\pm$2.42) | **64.69**($\pm$1.55) | **100.00**($\pm$0.00) | **100.00**($\pm$0.00) | **96.95**($\pm$0.18) | **99.99**($\pm$0.01) | **86.40**($\pm$8.63) | **99.96**($\pm$0.08) | **62.78**($\pm$6.95) | **88.06**($\pm$1.66) |
| GT | CTFG-GT | **95.80**($\pm$4.57) | **99.26**($\pm$1.12) | **51.81**($\pm$2.97) | **63.26**($\pm$2.73) | **100.00**($\pm$0.00) | **100.00**($\pm$0.00) | **96.57**($\pm$0.80) | **99.94**($\pm$0.18) | **76.80**($\pm$9.66) | **99.03**($\pm$0.98) | **65.17**($\pm$8.11) | **81.12**($\pm$1.03) |

### 5.5 Objective Function for Representational Consistency of Identical Transformations

The symmetry loss below enforces relative consistency between a given pair $(x_i, x_j)$, but the induced latent transformation still depends on the specific pair, which is precisely the pair-dependency issue in Definition 3.1. To eliminate this dependency, we additionally introduce a learnable canonical reference point $\theta_c$ that anchors each sample to an absolute position in latent space independently of which pair induced the transformation.

**Symmetry Loss (Relative Position).** As we define the group action $act(g^c, \theta_i)$ and step size $k'$, we implement the symmetry loss $\mathcal{L}_s$ to satisfy $\theta_j = \frac{2\pi}{N} k'_{i \to j} + \theta_i$:

$$\mathcal{L}_s = ||f' \circ f \circ q_\phi(x_j) - \left(\frac{2\pi}{N} k'_{i \to j} + f' \circ f \circ q_\phi(x_i)\right)||_2^2. \tag{6}$$

We set the code loss $\mathcal{L}_{code} = \mathcal{L}_{grid} + \mathcal{L}_s$.

**Canonical Loss (Absolute Position)** The defined symmetry $g^c$ represents the movement between two latent vectors ($\theta_i$ and $\theta_j$). It implies that learning symmetries depends on pairs of observations. To eliminate this dependency on specific observations, we propose learning absolute transformations through a learnable canonical point $\theta_c$ to satisfy $\theta_i = \frac{2\pi}{N} k'_{c \to i} + \theta_c$:

$$\mathcal{L}_c = ||f' \circ f \circ q_\phi(x_i) - \left(\frac{2\pi}{N} k'_{c \to i} + \theta_c\right)||_1, \tag{7}$$

where $\theta_c$ is a learnable canonical point $\theta_c \in \Theta^n$, $k'_{c \to i} = l_i$, and $|| \cdot ||_1$ is a L1 norm.

**Objective Loss.** Our objective losses are defined as 1) $\mathcal{L}_{GT} = \mathcal{L}_{recont} + \alpha \mathcal{L}_{code} + \gamma \mathcal{L}_c$ for Cayley transform and fixed grid Ground Truth model (CTFG-GT), 2) $\mathcal{L}_{Sup} = \mathcal{L}_{GT} + \beta \mathcal{L}_{pred}$ for supervised method (CTFG-SP), and 3) $\mathcal{L}_{weak-Sup} = \mathcal{L}_{Sup} + \lambda \mathcal{L}_{pl}$ for weakly-supervised method (CTFG-wSP).

## 6 Experiments

**Common Datasets.** We utilize the dSprites (Matthey et al., 2017), 3D Shapes (Burgess & Kim, 2018), and MPI3D (Gondal et al., 2019) datasets for compositional generalization and disentanglement learning tasks. More details are in Appendix D.2.

### 6.1 Disentanglement Learning

**Settings.** We set the common hyper-parameters of the proposed method $\alpha \in \{100, 1000\}$, $\gamma = 1$ for supervised and ground truth model, $\beta \in \{1.0, 2.0\}$ for supervised method, and $\lambda = 1.0, p = 0.5$ for weakly-supervised method. We run 10 seed variance over each model with seed $\in \{1, 2, \ldots, 10\}$. More details are in Appendix D.4.

**Case Studies: Do Any Ground Truth Based Methods Encourage Representational Consistency of Identical Transformations?** As shown in Section 4, we briefly show the difficulty of prior symmetry representations ($GL(n)$, vector addition, and subjective function) for the disentangled representation following

ground truth based method in Section 5.4. We demonstrate that previous methods are limited in preserving the dataset's symmetry and factor information. Consequently, we have adopted these three types of symmetry instead of our method. As indicated in Table 6, the CTFG-GT method outperforms other methods across all metrics. This suggests that enforcing the representational consistency of identical transformations in the angle space is a more suitable method for disentanglement learning.

**Quantitative Results.** Although comparisons with unsupervised methods should be interpreted cautiously due to the different supervision regimes, CTFG-SP and CTFG-GT achieve consistently strong performance on datasets, providing empirical reference points for the representational capacity of the framework, as shown in Table 3. In particular, they achieve near-saturated disentanglement scores on 3D Shapes and consistently strong performance on dSprites and MPI3D, indicating that the proposed framework remains effective under stronger supervision in these benchmark settings.

More importantly, under a direct weakly supervised comparison, CTFG-wSP generally outperforms Ada-GVAE, achieving higher scores on 11 out of 12 dataset-metric pairs. These results support the effectiveness of our method in learning disentangled representations under comparable weak supervision.

**Disentanglement vs. Reconstruction.** Most disentangled representation learning models face a trade-off between reconstruction error and disentanglement metrics (Kingma & Welling, 2013; Chen et al., 2018; Kim & Mnih, 2018; Zhu et al., 2021; Keurti et al., 2023; Higgins et al., 2017; Locatello et al., 2020a). However, our results suggest that this trade-off can be mitigated in the evaluated setting, as illustrated in Figure 4. Although our model's reconstruction error is two times lower than the baselines, it achieves higher disentanglement performance than the others with the MPI3D datasets. Further details are provided in Appendix F.1.

**Direct Measurement of Symmetry Consistency.** To directly validate Definition 3.1, we measure the cosine similarity between latent differences induced by identical ground-truth transformations. Specifically, for 50,000 sample pairs satisfying $l_1 - l_2 = l_3 - l_4$ under ground-truth labels, we compute $\cos(\boldsymbol{\theta}_1 - \boldsymbol{\theta}_2, \boldsymbol{\theta}_3 - \boldsymbol{\theta}_4)$ over 10 random

Table 4: Symmetry consistency scores across datasets.

|  | CTFG-sWP | CTFG-SP | CTFG-GT |
|---|---|---|---|
| dSprites | 0.9693(±0.0002) | 0.9889(±0.0001) | 1.0000(±0.0000) |
| 3D Shapes | 0.9835(±0.0002) | 0.9982(±0.0001) | 1.0000(±0.0000) |
| MPI3D | 0.9853(±0.0001) | 0.9984(±0.0001) | 1.0000(±0.0000) |

seeds. As shown in Table 4, all CTFG variants achieve near-perfect consistency across all datasets: CTFG-GT attains 1.0000 on every benchmark, while CTFG-SP and CTFG-wSP reach above 0.99 and 0.96, respectively. These results confirm that the proposed framework produces stable latent representations of identical transformations across different sample pairs, directly supporting the claim of Definition 3.1.

**Qualitative Analysis.** As shown in Figure 5a–5f, the baseline results show that multiple factors are changed when a single dimension value is changed on both the 3D Shapes and MPI3D datasets. Also, objects disappear at certain intervals in the baseline results. On the other hand, CTFG-SP and CTFG-GT show better results than the baselines. Compared to the baseline model, the cases of overlapping factors in a single dimension are reduced by the proposed models on both the 3D Shapes and MPI3D datasets.

## 6.2 Compositional Generalization

As shown in Section 6.1, encouraging a representational consistency of identical transformations leads to improved disentanglement performance in controlled settings. Because an important goal of disentangled

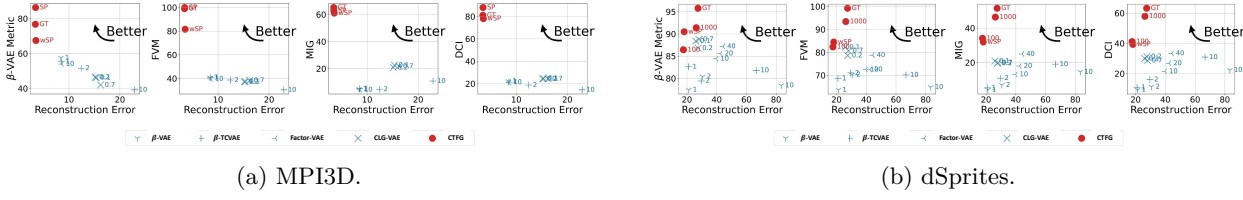

(a) MPI3D.  (b) dSprites.

Figure 4: Reconstruction error vs. evaluation metrics of the MPI3D and dSprites dataset ($\beta$-VAE metric, FVM, MIG, and DCI). The top left side indicates the best results on both objectives

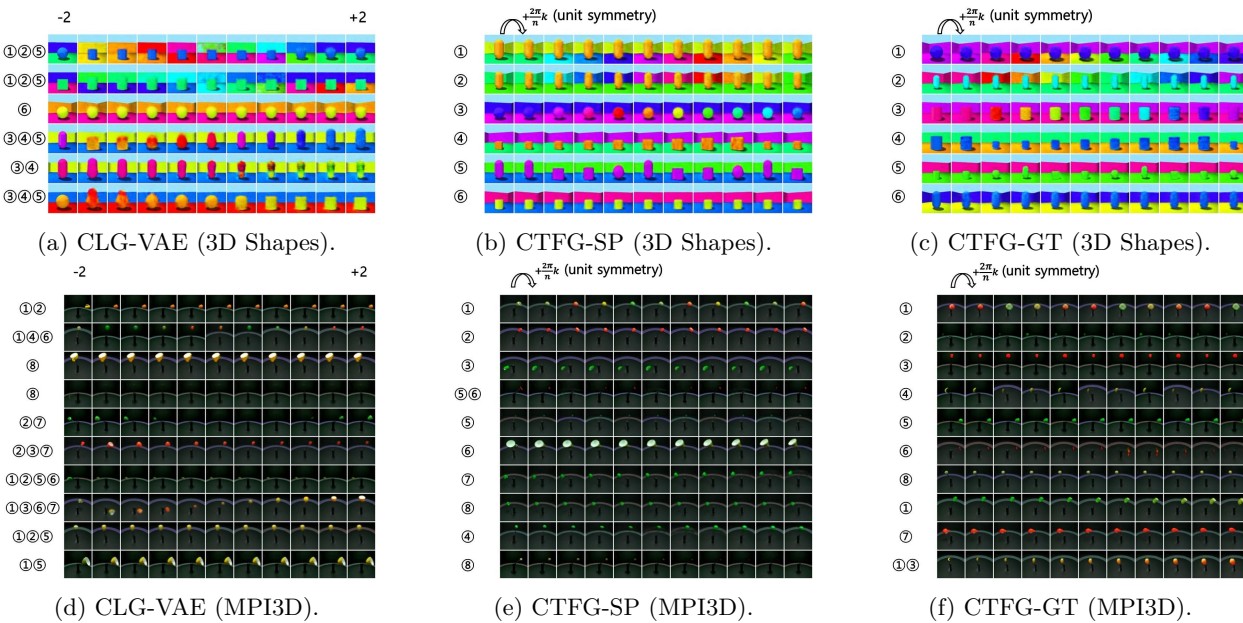

Figure 5: Alignment of a factor and a dimension: How many factors are changed following the dimension (column-wise direction)? The $1^{st}$ column images are randomly selected from the dataset. Each row indicates each dimension of each model. The Commutative Lie Group VAE trace each dimension value from -2 to +2. The proposed methods apply a group action $+\frac{2\pi}{n}$ to the selected images a total of 10 times. The numbers in Figure 5a-5c refer to factors of the 3D Shapes dataset: ①, ②, ③, ④, ⑤, and ⑥ refer to floor color, wall color, object color, scale, shape, and orientation, respectively. The numbers in Figure 5d-5f refer to factors of the MPI3D dataset: ①, ②, ③, ④, ⑤, ⑥, and ⑦ refer to object color, object shape, object size, camera height, background color, horizontal axis, and vertical axis, respectively.

representations is to support generalization, we further evaluate our approach in compositional generalization settings.

**Settings.** We excess Recombination-to-Element (R2E) and the Recombination-to-Range (R2R) tasks. We separate the training and test datasets following previous studies (Montero et al., 2021; Hwang et al., 2023), with additional details and hyper-parameter tuning provided in Appendix D.3. We run each model with three seeds $\in \{1, 2, 3\}$. We assess the reconstruction error, a general compositional generalization metric.

**Quantitative Results.** As shown in Table 5, the proposed method CTFG-GT is significantly improved with all datasets. It implies that the representational consistency of identical transformations also impacts to compositional generalization. Also, the supervised method CTFG-SP demonstrates advancements in all

Table 5: Compositional generalization performance of dSprites, 3D Shapes, and MPI3D datasets. We select the best results from the hyper-parameter tunings. The evaluation metric is the reconstruction error (BCE loss for dSprites, and MSE loss for 3D Shapes and MPI3D).

|  | Method | dSprites | | 3D Shapes | | MPI3D | |
|---|---|---|---|---|---|---|---|
|  |  | R2E | R2R | R2E | R2R | R2E | R2R |
| Base | $\beta$-VAE | 10.85(±0.67) | 179.52(±12.15) | 16.59(±1.72) | 268.59(±76.59) | 6.63(±0.65) | 8.50(±0.55) |
|  | $\beta$-TCVAE | 10.73(±0.03) | 153.75(±7.65) | 14.74(±0.14) | 221.72(±41.57) | 5.60(±0.21) | 8.73(±0.36) |
| Equiv. | CLG-VAE | 10.30(±0.20) | 246.37(±53.42) | 21.86(±0.98) | 276.81(±17.01) | 11.68(±0.85) | 18.36(±1.08) |
|  | VAE-MAGA | 11.22(±0.48) | 178.39(±11.64) | 18.84(±3.32) | 213.26(±41.76) | 5.43(±0.59) | 8.44(±0.44) |
|  | CTFG-wSP | **10.15**(±0.56) | **133.50**(±5.72) | **13.38**(±1.32) | **165.96**(±7.74) | **4.38**(±0.14) | **7.64**(±0.01) |
|  | CTFG-SP | **7.24**(±0.94) | **135.70**(±16.48) | **10.23**(±0.67) | **108.44**(±5.82) | **2.92**(±0.03) | **4.16**(±0.14) |
|  | CTFG-GT | **8.56**(±0.40) | **123.02**(±10.84) | **9.29**(±0.34) | **114.89**(±11.80) | **2.56**(±0.19) | **5.26**(±0.25) |

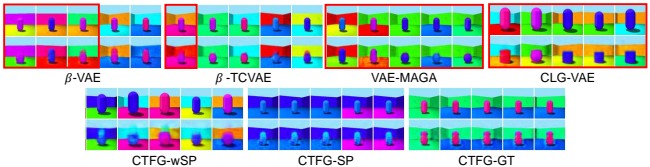

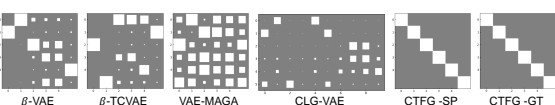

(a) Visualization of generated images of the worst 5 samples of the R2R task. Each $1^{st}$ and $2^{nd}$ row of models shows the group truth samples and the generated results, respectively. The red box indicates the negative results, which do not contain all the semantics of the ground truth. We utilize randomly selected pivot images as introduced in Hwang et al. (2023) for the CTFG model.

(b) Visualization of DCI metric of the 3D Shapes dataset (training set). A more sparse matrix implies a better disentangled representation. The x-axis refers to the index of the latent vector. The y-axis represents the factor of the dataset, from 1 to 6, corresponding to floor hue, wall hue, object hue, scale, shape, and orientation.

Figure 6: Qualitative results of compositional generalization of 3D Shapes dataset.

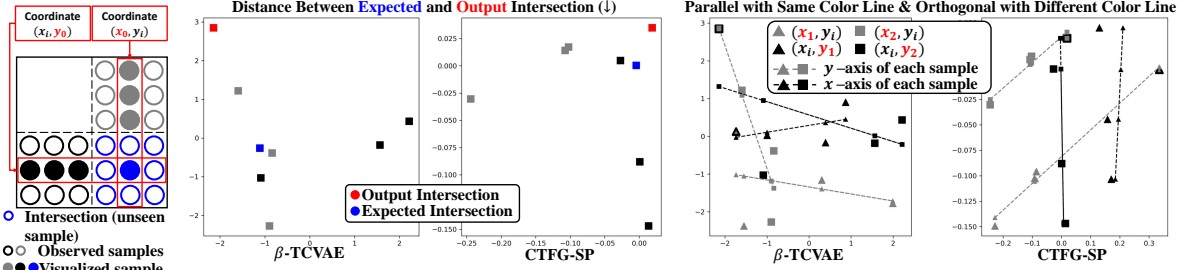

(a) Dataset Composition  (b) Consistency of Transformations  (c) Factor alignment over latent vector dimension

Figure 7: Alignment between factors and latent vector dimensions. As illustrated in (a), let the coordinates of the black points be denoted as $(x_i, y_0)$ and those of the gray points as $(x_0, y_i)$. The blue point then corresponds to the coordinate $(x_0, y_0)$, representing the intersection of independently observed components. 1) This setup captures the core intuition behind compositional generalization: novel factors can emerge at the intersection of separately observed attributes. We set black and gray points as $(F_i, F_0')$ and $(F_0, F_i')$, respectively. Then we extend this core intuition to the latent vector space to demonstrate how each model follows this intuition and visualize it through (b). 2) Since all points are defined by their $(x, y)$ coordinates, points with the same $x$-coordinate align along the $x$-axis, and those with the same $y$-coordinate align along the $y$-axis. Extending this idea to the latent vector space, if two samples share the same factor of variation, their latent vectors are expected to align along the axis corresponding to that factor. We visualize this core intuition in (c) to assess how well each model adheres to this structural alignment.

datasets, and the weakly supervised approach, CTFG-wSP, achieves performance close to the CTFG-GT and CTFG-SP on the challenging R2R task and across complex datasets, as shown in Table 5.

**Qualitative Results.** As illustrated in Figure 6a, both proposed methods preserve the semantics of the ground truth while baselines struggle to retain the semantics of unseen samples (ground truth). Comparing the VAE-MAGA and our models, forcing the consistency of the transformation method has a much greater impact on generalization. Additionally, our models exhibit a disentangled representation compared to the baselines, as shown in Figure 6b. This implies that a disentangled representation incorporating the symmetry structure promotes compositional generalization. Details of other results on the dSprites and MPI3D dataset are provided in Appendix E.

**Alignment of Unseen Samples.** First, as shown in Figure 7b, the baseline model yields a latent vector (red dot) that is substantially distant from the expected intersection (blue dot). In contrast, as shown in Figure 7b, the latent vector produced by our model (CTFG) is positioned much closer to the intersection, clearly outperforming the baseline. These findings suggest that our method learns a more factorized latent representation, where each dimension more effectively captures a specific factor of variation. As a result, novel combinations can be composed more meaningfully within the latent space. Second, As shown in Figure 7c,

Table 6: Ablation study results on the dSprites dataset. We isolate each component from the full CTFG-SP model shown in Table 3. The $S^2/S$-$\beta$-VAE baseline is trained with factor-level supervision, while CTFG-SP variants use transformation-level consistency supervision. FG denotes the fixed-grid component.

| Method | Inductive Bias | | | dSprites | | | | | | |
|---|---|---|---|---|---|---|---|---|---|---|
| | Supervision | Architecture | RCIT | beta-VAE | FVM | MIG | DCI | | R2E | R2R |
| $S^2/S$-$\beta$-VAE | ✓ | ✗ | ✗ | 75.71($\pm$11.16) | 70.34($\pm$14.01) | 7.59($\pm$5.80) | 11.52($\pm$6.14) | | 10.54($\pm$0.68) | 164.29($\pm$20.23) |
| CTFG-SP (w/o FG) | ✓ | ✗ | ✓ | 90.75($\pm$5.23) | 92.36($\pm$2.54) | 49.06($\pm$5.46) | 62.55($\pm$6.55) | | 8.87($\pm$0.91) | 138.80($\pm$17.19) |
| CTFG-SP (w/o $\mathcal{L}_c$) | ✓ | ✓ | △ | 90.33($\pm$4.00) | 92.63($\pm$3.17) | 50.29($\pm$2.55) | 62.52($\pm$3.25) | | 8.71($\pm$0.97) | 151.30($\pm$19.09) |
| CTFG-SP (w/o $\mathcal{L}_s$) | ✓ | ✓ | △ | 67.75($\pm$7.67) | 58.02($\pm$6.85) | 3.02($\pm$0.86) | 6.47($\pm$1.50) | | 9.02($\pm$1.12) | 159.24($\pm$23.51) |
| CTFG-SP | ✓ | ✓ | ✓ | **91.40**($\pm$4.99) | **93.74**($\pm$1.82) | **51.02**($\pm$2.42) | **64.69**($\pm$1.55) | | **7.24**($\pm$0.94) | **135.70**($\pm$16.48) |

the baseline fails to produce latent axes that are aligned across samples sharing the same factor (same color dashed lines are not parallel), and the axes do not exhibit orthogonality with respect to differing factors (different color dashed lines are not orthogonal). In contrast, our model yields latent vectors whose axes corresponding to shared factors are more consistently aligned, exhibiting a more structured and factorized representation compared to the baseline, as shown in Figure 7c.

**Stability of Generalization.** As illustrated in Figure 8, the baselines often show unstable generalization behavior during training on dSprites and 3D Shapes, whereas our model exhibits more stable improvement over training. These observations are consistent with the view that inductive bias is helpful for generalization in this setting, and further suggest that representational consistency of identical transformations contributes to improve generalization behavior.

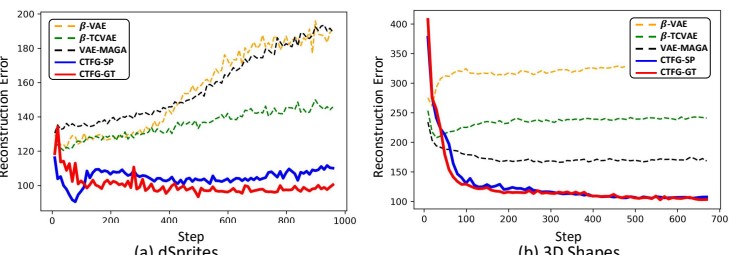

Figure 8: Test error during training.

### 6.3 (W 1, MA 1, MI 2 & S 3-2, MI 1) Discussion

**Impact of RCIT Supervision.** As shown in Section 3.2 and Table 6, the comparison between $S^2/S$-$\beta$-VAE Locatello et al. (2020b) and CTFG-SP (w/o FG) provides quantitative evidence supporting the need for representational consistency of identical transformations in disentanglement learning.

**Impact of Fixed Grid.** we evaluate CTFG-SP (w/o FG) as a continuous latent variant that retains consistency supervision. Removing the fixed grid causes a small performance decrease compared with the full model as shown in Table 6. CTFG-SP (w/o FG) also substantially outperforms the supervised $S^2/S$-$\beta$-VAE baseline across all disentanglement metrics. These results show that the fixed grid contributes to performance, but does not alone account for the observed improvement.

**Impact of Symmetry Loss and Canonical Loss.** As shown in Table 6, removing the canonical consistency loss $\mathcal{L}_c$ results in marginal degradation across the disentanglement metrics. This indicates that $\mathcal{L}_c$ provides auxiliary support and does not independently explain the improved performance. In contrast, removing the relative consistency loss $\mathcal{L}_s$ causes a much larger degradation. The $\beta$-VAE, FVM, MIG, and DCI scores decrease to 67.75, 58.02, 3.02, and 6.47, respectively. This result demonstrates that learning consistent relative transformations is closely related to the quality of the learned disentangled representation.

Overall, the ablation results show that the improvement cannot be explained solely by fixed-grid quantization or the canonical anchor loss, while the large degradation without $\mathcal{L}s$ highlights the importance of relative consistency. These results support representational consistency as a useful inductive bias in the considered settings.

**Empirical Motivation for RCIT in Compositional Generalization** : An identical semantic transformation in factor space acts as a reusable rule independently of the source state. RCIT transfers this property to the latent space, allowing the same latent action to be reused for unseen compositions. Under a

controlled comparison using the same factor-label information without the fixed grid, CTFG-SP (w/o FG) reduces the R2E and R2R errors of $S^2/S$-$\beta$-VAE from 10.54 and 164.29 to 8.87 and 138.80, respectively. The full CTFG-SP further reduces them to 7.24 and 135.70, supporting RCIT as a useful inductive bias for compositional generalization.

**(MI 1)** **Loss selection between L1 and L2.** Table 7 compares the L1 and L2 losses for the relative consistency loss $\mathcal{L}_s$ and the canonical consistency loss $\mathcal{L}_c$. Using L2 for $\mathcal{L}_s$ and L1 for $\mathcal{L}_c$ achieves the highest mean $\beta$-VAE, FVM, and DCI scores. Although the L1–L1 configuration obtains a slightly higher MIG score, the selected L2–L1 configuration provides the best overall performance across the evaluated metrics. We therefore use L2 for the relative consistency objective ($\mathcal{L}_s$) and L1 for the canonical objective ($\mathcal{L}_c$).

## 7 Related Works

**Compositional Generalization.** Recent research in compositional generalization shows that models trained on disentanglement learning and verified through ground truth experimentally demonstrate that high disentangled representation does not necessarily imply compositional generalization (Montero et al., 2021; 2022; Schott et al., 2022). Differently, we consider the symmetry-based disentangled representations, recently studied in

Table 7: Ablation study on the loss types for $\mathcal{L}_s$ and $\mathcal{L}_c$.

| $\mathcal{L}_s$ | $\mathcal{L}_c$ | $\beta$-VAE | FVM | MIG | DCI |
|---|---|---|---|---|---|
| L1 | L1 | $90.00_{\pm2.58}$ | $92.91_{\pm0.19}$ | $\mathbf{51.92_{\pm2.40}}$ | $57.81_{\pm0.92}$ |
| L1 | L2 | $90.67_{\pm2.31}$ | $90.75_{\pm0.14}$ | $51.24_{\pm2.12}$ | $57.29_{\pm0.35}$ |
| **L2** | **L1** | $\mathbf{91.40_{\pm4.99}}$ | $\mathbf{93.74_{\pm1.82}}$ | $51.02_{\pm2.42}$ | $\mathbf{64.69_{\pm1.55}}$ |
| L2 | L2 | $88.00_{\pm4.90}$ | $92.19_{\pm3.70}$ | $40.92_{\pm3.36}$ | $53.67_{\pm1.58}$ |

the disentanglement learning field to preserve the symmetry structure of the dataset in latent vector space. MAGANet (Hwang et al., 2023) dramatically improved compositional generalization performance by learning symmetries with the Glow model. In contrast, we study how a VAE-based model can be designed to promote more consistent representations of identical transformations.

**Disentanglement Learning.** The initially proposed methods, such as Chen et al. (2018); Kim & Mnih (2018); Higgins et al. (2017), partition each dimension to ensure mutual exclusivity, employing measures like mutual information or total correlation. However, these approaches do not account for the symmetry structure of the dataset space. Defining the symmetry group and group action as a General Linear group and matrix multiplication (Zhu et al., 2021; Jung et al., 2024; Marchetti et al., 2023) enhances disentanglement performance. Nevertheless, we theoretically demonstrate the limitations of the General Linear group for cyclic semantics in the disentangled space. Other works commonly define the symmetry group acting on the latent vector space as a cyclic group with surjective functions (Yang et al., 2021; Keurti et al., 2023; Falorsi et al., 2018). Differently, our focus is on employing isomorphism to represent the cyclic group rather than a homomorphism.

## 8 Conclusion

In this paper, we study the problem of representing identical semantic changes more consistently for disentangled representations. We analyze three commonly used group settings that can make such consistency difficult to maintain in disentangled latent spaces, and propose a Cayley-transform-based cyclic representation together with a practical learning framework. In controlled benchmark settings, the weakly supervised variant achieves strong disentanglement and compositional generalization, while the supervised variants further demonstrate the potential of the proposed principle. **(W 3)** We believe this study offers a meaningful step toward understanding how representational consistency of identical transformations can serve as a principled inductive bias for disentanglement learning and compositional generalization. While our analysis focuses on cyclic factors and therefore does not cover all variations in real-world data, future work will extend RCIT to broader symmetry groups and more diverse real-world factors of variation.

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

Table 8: Notation Table

| Set | | | | | |
|---|---|---|---|---|---|
| $F$ | Set of latent factors | $X$ | Dataset | $Z$ | Set of latent vectors |
| $\mathcal{F}$ | Space of latent factors | $\mathcal{X}$ | Space of datasets | $\mathcal{Z}$ | Space of latent vectors |
| $\mathcal{Y}$ | Space of latent vectors | $\Theta$ | Set of angles: $\{\theta \mid -\pi < \theta \leq \pi\}$ | | |
| **Group** | | | | | |
| $G$ | Group | $G^F$ | Group acted on the set of latent factors | $G^z$ | Group acted on the set of latent vectors |
| $G^c$ | Cyclic group | $g$ | Group element of $G$ | $\mathfrak{g}$ | Lie algebra of $GL(n,\mathbb{R})$ |
| $act(\cdot,\cdot)$ | Group action | $\circ_F$ | Binary operation of $G_F$ | $\circ_z$ | Binary operation of $G_z$ |
| $GL(n)$ | General Linear group | $GL'(n)$ | General Linear group implemented by matrix exponential $GL'(n) \subset GL(n)$ | $g_{z_j}^{(i)\to(i+1)}$ | Symmetry between $z_j^i$ and $z_j^{i+1}$ |
| $S^1$ | Circle group: $\{z \in \mathbb{C} : |z| = 1\}$ | $\mathbb{R}/\mathbb{Z}$ | $\{r + \mathbb{Z} \mid r \in \mathbb{R}\}$ | $G \cong H$ | $G$ and $H$ are isomorphic |
| **Function** | | | | | |
| $\Omega$ | $\mathcal{F} \to \mathcal{X}$ | $q_\phi$ | $\mathcal{X} \to \mathcal{Z}$ | $\Gamma$ | $G_F \to G_z$ |
| $\Gamma'$ | $G_z \to G_y$ | $e^x$ | Matrix exponential | $f$ | $\mathbb{R} \to \{c \mid c \in \mathbb{C}, |c| = 1\}$ |
| $f'$ | $\{c \mid c \in \mathbb{C}, |c| = 1\} \to \Theta$ | $C.E.(\cdot)$ | Cross-entropy loss | $D_{\mathrm{KL}}(\cdot\|\cdot)$ | Kullback–Leibler divergence |
| $\mathcal{N}(\boldsymbol{\mu}, \boldsymbol{\Sigma})$ | Gaussian distribution | $\mathcal{U}\{a,b\}$ | Discrete uniform distribution | | |
| **Linear Algebra** | | | | | |
| $\boldsymbol{I}$ | Identity matrix | $\boldsymbol{V}$ | Codebook | $\boldsymbol{J}$ | Jordan normal form |
| $\boldsymbol{N}$ | Nilpotent matrix | $\boldsymbol{M}^k$ | Zeros matrix except $k^{\text{th}}$ column vector | $\boldsymbol{m}^k$ | $k^{\text{th}}$ column vector of $\boldsymbol{M}^k$ |
| $\boldsymbol{z}$ | Latent vector $\in \mathbb{R}^n$ | $\boldsymbol{z}^k$ | $k^{\text{th}}$ dimension value of $\boldsymbol{z}$ | $\boldsymbol{c}$ | Latent vector $\in \mathbb{C}^n$ |
| $\boldsymbol{c}^k$ | $k^{\text{th}}$ dimension value of $\boldsymbol{c}$ | $\boldsymbol{\theta}$ | $\boldsymbol{\theta} \in \Theta^n$ | $\boldsymbol{\theta}^k$ | $k^{\text{th}}$ dimension value of $\boldsymbol{\theta}$ |
| $\theta_c$ | Canonical point | $\vec{0}$ | Zeros vector | $\Delta\boldsymbol{v}$ | $\sum_k \Delta\boldsymbol{v}^k$ |
| $\Delta\boldsymbol{v}^k$ | Sparse vector ($k^{\text{th}}$ dim. value $\in \mathbb{R} \setminus \{0\}$) | $\boldsymbol{k}$ | Step | $\boldsymbol{k}_{i\to j}$ | Step from $\boldsymbol{\theta}_i$ to $\boldsymbol{\theta}_j$ |
| **Others** | | | | | |
| $\mathbb{R}$ | Real number | $\mathbb{Z}^+$ | Integer | $\mathbb{C}$ | Complex number |
| $\mathfrak{R}$ | Real part of complex number | $\mathfrak{I}$ | Imaginary part of complex number | $|F_i|$ | Number of factors |
| $\|\cdot\|$ | L1 norm | $\|\cdot\|_2$ | L2 norm | $l$ | Ground truth |

# A  Details of Preliminaries

**Normal Subgroup:** A subgroup $N$ of a group $G$ is termed a normal subgroup of $G$ if and only if $gng^{-1} \in N$ for all $g \in G$ and $n \in N$. Equivalently, a normal subgroup is one that is invariant under conjugation.

**Cosets:**  Given a subset $H$ of a group $G$, the left coset of $H$ with respect to an element $g \in G$ defined as: $gH = \{gh : h \in H\}\forall g \in G$, and the right coset $Hg = \{hg : h \in H\}\forall g \in G$.

**Factor (Quotient) Group:**  If $N$ is a normal subgroup of group $G$, the factor group (or quotient group) $G/N$ to be the set of all left cosets of $N$ in $G$: $G/N = \{aN : a \in G\}$.

# B  Proof of Limitations

We consider the dimension-wise disentangled representation, so we constraint a few conditions as follows:

**Condition B.1.** There exists an equivariant function $q_\phi \circ \Omega : \mathcal{F} \to \mathcal{Z}$ mapping fully disentangled factor and latent space.

**Condition B.2.** $\mathcal{Z}$ is a $G_z$-set that is a symmetry group acting on $Z$.

**Condition B.3.** Group element $g_z$ only affects to a single dimension value of latent vector $\boldsymbol{z}$, where $g_z \in G_z$.

## B.1  Summary

**Case 1: A Limitation of $GL(n)$**  General Linear group $GL(n)$, used in prior works (Zhu et al., 2021; Kuzina et al., 2022; Miyato et al., 2022; Marchetti et al., 2023), is limited in representing the representational consistency of identical transformations for disentangled representation. We first show the property of disentangled representation with matrix exponential. Then we show the limitation of $GL(n)$.

**Proposition B.4.** *Let the symmetry group $G_z$ ($GL'(n)$) is defined as a subgroup of the General Linear group that implemented with matrix exponential, where $GL'(n) = \{e^{\boldsymbol{M}}|\boldsymbol{M} \in \mathbb{R}^{n\times n}\}$, $g^k$ is an element of $GL'(n)$, and $g = \prod_k g^k$. Then $e^g\boldsymbol{z} = e\boldsymbol{I}g\boldsymbol{z} + \boldsymbol{v}'$.*

**Theorem B.5.** *(Limitation of $GL'(n)$) By Proposition B.4, if $g \in GL'(n)$ is compatible with Conditions 4.1– 4.3 and, $|G^{F^i}| > 2$, then $g = \boldsymbol{I}$.*

**Theorem B.6.** *(Limitation of $GL(n)$) If Conditions 4.1–4.3 hold, $|G^{F^i}| > 2$, and $H = \{h \in GL(n) \mid h = \boldsymbol{I} + \boldsymbol{M}^i\}$, then $h = \boldsymbol{I}$ is the only element of $H$.*

Therefore, the limitation of $GL(n)$ is that only the $\boldsymbol{I}$ represents the representational consistency of identical transformations with disentangled representation according to the Theorem Theorem B.5, and Theorem B.6. It implies that if $g \neq \boldsymbol{I}$, then $g$ can not represent the representational consistency of identical transformations. More details are in Appendix Section B.2.

**Case 2: A Limitation of Using Vector Addition**   Another setting that causes problems in maintaining representational consistency of identical transformations for cyclic group using vector addition is utilized to define a group action between two latent vectors for an equivariant model, as used in Hwang et al. (2023); Balabin et al. (2024).

**Corollary B.7.** *If the group action is defined as $act(g, \boldsymbol{z}_i) = g + \boldsymbol{z}_i$, then only zero vector represent the representational consistency of identical transformations for cyclic group with disentangled representation, where $\boldsymbol{z} \in \mathbb{R}^n$.*

According to the Theorem Theorem B.7, it also shows a limitation in that only the identity element $\vec{0}$ represents the consistent symmetry. More details of the proof are in Appendix Section B.3.

**Case 3: A Limitation of Surjective Function**   The last setting is a surjective function that maps latent vectors to the unit circle (Yang et al., 2021; Tonnaer et al., 2022; Cha & Thiyagalingam, 2023), causing undifferentiated symmetries under more general latent factors of variation and losing part of the dataset's symmetry information.

**Corollary B.8.** *By the equivariant and surjective function $b : \mathcal{Z} \to \mathcal{Y}$, the capacity of $\mathcal{Z}$ and $\mathcal{Y}$ is $|\mathcal{Z}| \geq |\mathcal{Y}|$ then $\Gamma'$ is an endomorphism because $|G_{\mathcal{Z}}| \geq |G_{\mathcal{Y}}|$. On the other hand, isomorphism identically maps the two spaces $(|G_{\mathcal{Z}}| = |G_{\mathcal{Y}}|)$.*

Therefore, if $b$ is surjective and not injective, then there exists at least one case where $\Gamma'(g_i) = \Gamma'(g_j)$. It implies that loss of symmetry structure occurs with a surjective function. More details of the proof are in Appendix Section B.4.

## B.2   Proof: Limitations of $GL(n)$

**Proof: Limitations of $GL(n)$, implemented by the matrix exponential, to Represent the Cyclic Group on the Disentangled Space**

**Condition B.9.** The symmetry group $G_z$ ($GL'(n)$) acting on latent vector space is defined as a subgroup of the General Linear group, implemented with matrix exponential.

**Condition B.10.** For $g^k \in \mathbb{R}^{n \times n}$ and $g = \prod_k g^k$, $g^k$ only affects the $k^{th}$ dimension value of vector $\boldsymbol{z}$.

In prior works (Jung et al., 2024; Zhu et al., 2021; Kuzina et al., 2022; Miyato et al., 2022; Marchetti et al., 2023), the General Linear group $GL(n)$ is usually implemented with the Lie algebra $\mathfrak{g}$ to represent the symmetries between two inputs in the latent vector space:

$$g = e^{\sum_i \alpha_i \mathfrak{g}_i}, \tag{8}$$

where $g \in GL(n)$, $\alpha \in \mathbb{R}$, $\mathfrak{g} \in \mathbb{R}^{n \times n}$, and the matrix exponential $e^{\mathfrak{g}}$ defined as $e^{\mathfrak{g}} = \sum_{n=0}^{\infty} \frac{1}{n!} \mathfrak{g}^n$. In addition, group $GL(n)$ acts on the latent vector space $\mathcal{Z}$ with group action:

$$act(g, \boldsymbol{z}) = g\boldsymbol{z}, \tag{9}$$

where latent vector $\boldsymbol{z} \in \mathbb{R}^n$, commonly used in previous works (Zhu et al., 2021; Jung et al., 2024; Kuzina et al., 2022; Marchetti et al., 2023). We first show the property of disentangled representation with matrix exponential.

**Proposition B.11.** *Let the symmetry group $G_z$ $(GL'(n))$ is defined as a subgroup of the General Linear group that implemented with matrix exponential, where $GL'(n) = \{e^{\boldsymbol{M}} | \boldsymbol{M} \in \mathbb{R}^{n \times n}\}$, $g^k$ is an element of $GL'(n)$, and $g = \prod_k g^k$. Then $e^g \boldsymbol{z} = e\boldsymbol{I}g\boldsymbol{z} + \boldsymbol{v}'$.*

*Proof.* If group element $g$ acts on latent vector then, $g\boldsymbol{z} - \boldsymbol{z} = \Delta\boldsymbol{v}$, where $\Delta\boldsymbol{v} = \sum_k \Delta\boldsymbol{v}^k$, $\Delta\boldsymbol{v}^k$ is a sparse vector ($k^{th}$ dimension value $\in \mathbb{R}\backslash\{0\}$, otherwise it is zero), and $g^k \boldsymbol{z} - \boldsymbol{z} = \Delta\boldsymbol{v}^k$. Then we define $(g)^n \boldsymbol{z} - \boldsymbol{z} = n\Delta\boldsymbol{v} + (n-1)\boldsymbol{v}'_n$, where $\boldsymbol{v}'_n \in \mathbb{R}^n$ is an arbitrary real vector. Group element $g$ represents the cyclic semantics of the dataset space, then it satisfies the following equation:

$$
\begin{aligned}
(g_i - \boldsymbol{I})\boldsymbol{z} &= \Delta\boldsymbol{v} \\
\frac{1}{2!}((g_i)^2 - \boldsymbol{I})\boldsymbol{z} &= \frac{2}{2!}(\Delta\boldsymbol{v} + \frac{1}{2}\boldsymbol{v}'_2) \\
&\vdots \\
\lim_{n\to\infty} \frac{1}{n!}((g_i)^n - \boldsymbol{I})\boldsymbol{z} &= \lim_{n\to\infty} \frac{1}{(n-1)!}(\Delta\boldsymbol{v} + \frac{1}{n}\boldsymbol{v}'_n).
\end{aligned}
\tag{10}
$$

By adding left- and right-hand side of Eq. Equation (10), we then get:

$$
\begin{aligned}
&\Rightarrow (e^{g_i} - \boldsymbol{I})\boldsymbol{z} - (e-1)\boldsymbol{I}\boldsymbol{z} = e\boldsymbol{I}\Delta\boldsymbol{v} + \boldsymbol{v}' \\
&\Rightarrow e^{g_i}\boldsymbol{z} = e\boldsymbol{I}g_i\boldsymbol{z} + \boldsymbol{v}',
\end{aligned}
\tag{11}
$$

where $g_i \in G$, $\boldsymbol{v}' = \lim_n \sum_n \frac{1}{n!}\boldsymbol{v}'_n$ and $\boldsymbol{v}' = \boldsymbol{v}' + e\boldsymbol{I}\Delta\boldsymbol{v}$. $\qquad\square$

**Lemma B.12.** *By the Proposition Theorem B.11, if $\boldsymbol{v}' = \vec{0}$ in Eq. Equation (11), then $g = \boldsymbol{I}$ and the index of the nilpotent matrix of Jordan normal form of $\mathfrak{g}$ is 2.*

*Proof.* The trivial solution of $(e^g - e\boldsymbol{I}g)\boldsymbol{z} = 0, \forall \boldsymbol{z} \in \mathcal{Z}$ is that

$$
e^g - e\boldsymbol{I}g = 0.
\tag{12}
$$

Every matrix $\mathfrak{g} \in \mathbb{C}^{n \times n}$ has a Jordan normal form $\boldsymbol{J}$ as $\mathfrak{g} = \boldsymbol{S}\boldsymbol{J}\boldsymbol{S}^{-1}$. Then group element $(g = e^{\mathfrak{g}})$ follows as:

$$
\begin{aligned}
e^{\mathfrak{g}} &= \lim_{n\to\infty} \boldsymbol{I} + \boldsymbol{S}\boldsymbol{J}\boldsymbol{S}^{-1} + \frac{1}{2!}\boldsymbol{S}\boldsymbol{J}^2\boldsymbol{S}^{-1} + \cdots + \frac{1}{n!}\boldsymbol{S}\boldsymbol{J}^n\boldsymbol{S}^{-1} \\
&= \lim_{n\to\infty} \boldsymbol{I} + \boldsymbol{S}(\boldsymbol{J} + \frac{1}{2!}\boldsymbol{J}^2 + \cdots + \frac{1}{n!}\boldsymbol{J}^n)\boldsymbol{S}^{-1} \\
&= \boldsymbol{I} + \boldsymbol{S}(e^{\boldsymbol{J}} - \boldsymbol{I})\boldsymbol{S}^{-1} \\
&= \boldsymbol{S}e^{\boldsymbol{J}}\boldsymbol{S}^{-1}
\end{aligned}
\tag{13}
$$

In the same way, the exponential of $g$ is equal to:

$$
e^g = \boldsymbol{S}e^{e^{\boldsymbol{J}}}\boldsymbol{S}^{-1}
\tag{14}
$$

Therefore group element $g$ satisfies

$$
\begin{aligned}
\boldsymbol{S}e^{e^{\boldsymbol{J}}}\boldsymbol{S}^{-1} &= e\boldsymbol{I}\boldsymbol{S}e^{\boldsymbol{J}}\boldsymbol{S}^{-1} \\
\Rightarrow e^{e^{\boldsymbol{J}}} &= e^{\boldsymbol{I}}e^{\boldsymbol{J}} \ (\because e\boldsymbol{I} = e^{\boldsymbol{I}}) \\
\Rightarrow e^{e^{\boldsymbol{J}}} &= e^{\boldsymbol{I}+\boldsymbol{J}} \ (\because \boldsymbol{I}\boldsymbol{J} = \boldsymbol{J}\boldsymbol{I}) \\
\therefore e^{\boldsymbol{J}} &= \boldsymbol{I} + \boldsymbol{J}
\end{aligned}
\tag{15}
$$

If $\boldsymbol{J} = \boldsymbol{0}$, then $g$ satisfies the Eq. Equation (15) and $g = \boldsymbol{I}$. If $\boldsymbol{J} \neq \boldsymbol{0}$ then,

$$
e^{\boldsymbol{J}} = \begin{bmatrix} e^{\lambda_1} & e^{\boldsymbol{J}}_{1,2} & \cdots & e^{\boldsymbol{J}}_{1,n} \\ & e^{\lambda_2} & \cdots & e^{\boldsymbol{J}}_{2,n} \\ & & \ddots & \vdots \\ & & & e^{\lambda_n} \end{bmatrix}, \text{ and } \boldsymbol{I} + \boldsymbol{J} = \begin{bmatrix} \lambda_1 + 1 & (\boldsymbol{I}+\boldsymbol{J})_{1,2} & \cdots & (\boldsymbol{I}+\boldsymbol{J})_{1,n} \\ & \lambda_2 + 1 & \cdots & (\boldsymbol{I}+\boldsymbol{J})_{2,n} \\ & & \ddots & \vdots \\ & & & \lambda_n + 1 \end{bmatrix},
\tag{16}
$$

where empty values are all zero. To satisfy the Eq. Equation (15), $\lambda_i = 0$ for $e_i^\lambda = \lambda_i + 1$, then $\boldsymbol{J} = \boldsymbol{D} + \boldsymbol{N} = \boldsymbol{N}$ because diagonal of $\boldsymbol{D}$ $\lambda_i = 0$, where $\boldsymbol{D}$ is a diagonal matrix and $\boldsymbol{N}$ is a nilpotent matrix. Therefore,

$$
\begin{aligned}
e^{\boldsymbol{J}} &= e^{\boldsymbol{N}} \text{ and } \boldsymbol{I} + \boldsymbol{J} = \boldsymbol{I} + \boldsymbol{N} \\
\Rightarrow\ e^{\boldsymbol{N}} &= \boldsymbol{I} + \boldsymbol{N} \\
\Rightarrow\ \lim_{n \to \infty} \frac{1}{2!}\boldsymbol{N}^2 &+ \frac{1}{3!}\boldsymbol{N}^3 + \cdots + \frac{1}{n!}\boldsymbol{N}^n = 0
\end{aligned}
\tag{17}
$$

Therefore if the index of nilpotent matrix is 2 and $e_{i,j}^{\boldsymbol{J}} = (\boldsymbol{I} + \boldsymbol{J})_{i,j}$ where $i < j$, then it satisfies the Eq. Equation (12)

$\square$

**Lemma B.13.** *If the index of the nilpotent matrix of Jordan normal form of $\mathfrak{g}$ is 2, then $g = e^{\mathfrak{g}}$ does not represent the element of cyclic group.*

*Proof.* If $g$ is an element of cyclic group then there exists $n^{th}$ power of $g$ such that $g^n$ is the identity matrix.

$$
\begin{aligned}
g^n &= \boldsymbol{S}(\boldsymbol{I} + n\boldsymbol{N})\boldsymbol{S}^{-1} = \boldsymbol{I}\ (\because \boldsymbol{N}^2 = \boldsymbol{0}) \\
\Rightarrow\ \boldsymbol{I} + \boldsymbol{S}n\boldsymbol{N}\boldsymbol{S}^{-1} &= \boldsymbol{I} \\
\Rightarrow\ \boldsymbol{S}n\boldsymbol{N}\boldsymbol{S}^{-1} &= \boldsymbol{0}.
\end{aligned}
\tag{18}
$$

To satisfy the Eq. Equation (18), $\boldsymbol{N} = 0$ because the index of $\boldsymbol{N}$ is 2. $\square$

By the Lemma Theorem B.12 and Theorem B.13, there exists only one element to represent the element of cyclic group of the dataset in the disentangled space.

**Lemma B.14.** *If $\boldsymbol{v}' \in \mathbb{R}^n \backslash \{0\}$, then 1) $g^k$ represents the element of cyclic group as $g^k = \boldsymbol{I} + \boldsymbol{M}^k$ and $\boldsymbol{m}^k \in \mathbb{R}^n$, where $\boldsymbol{m}^k$ is a $k^{th}$ column vector of $\boldsymbol{M}^k$, $\boldsymbol{m}^j = 0$ and $j \in \{1, 2, \ldots, n\} \backslash \{k\}$.*

*Proof.* As we define that $g^k$ only affect to a single dimension value of $\boldsymbol{z}$ we rewrite Eq. Equation (11) as follows:

$$
e^{g^k} \boldsymbol{z} = e\boldsymbol{I}g^k \boldsymbol{z} + \boldsymbol{v}'^k,
\tag{19}
$$

where $\boldsymbol{v}'^k$ is a sparse vector. For Eq. Equation (19) $\forall \boldsymbol{z} \in \mathcal{Z}$ then symmetry $g^k$ follows as:

$$
e^{g^k} - e\boldsymbol{I}g^k = \begin{bmatrix} \vec{0} & \cdots & \vec{0} & \boldsymbol{m}^k & \vec{0} \cdots \vec{0} \end{bmatrix},
\tag{20}
$$

where $\vec{0}$ is a zero column vector.

Then, satisfying the Eq. Equation (20) and affecting a single dimension:

$$
\begin{aligned}
\boldsymbol{z}^{\mathsf{T}} g^k &= [z_1, \cdots, z_{k-1}, z_k + \alpha, z_{k+1}, \cdots, z_n] \\
\boldsymbol{z}^{\mathsf{T}} e\boldsymbol{I}g^k &= [ez_1, \cdots, ez_{k-1}, e(z_k + \alpha), ez_{k+1}, \cdots, ez_n] \\
\therefore\ \boldsymbol{z}^{\mathsf{T}} e^{g^k} &= [ez_1, \cdots, ez_{k-1}, z_k + \beta, ez_{k+1}, \cdots, ez_n] \\
\Rightarrow \boldsymbol{z}^{\mathsf{T}}(e^{g^k} - e\boldsymbol{I}) &= [0, \cdots, 0, (1 - e)z_k + \beta, \cdots, 0].
\end{aligned}
\tag{21}
$$

For Eq. Equation (21) for all $\boldsymbol{z}$, then

$$
\begin{aligned}
e^{g^k} - e\boldsymbol{I} &= [\vec{0} \cdots \vec{0}\ \boldsymbol{m}'^k\ \vec{0} \cdots \vec{0}] \\
\therefore e^{g^k} &= e\boldsymbol{I} + [\vec{0} \cdots \vec{0}\ \boldsymbol{m}'^k\ \vec{0} \cdots \vec{0}].
\end{aligned}
\tag{22}
$$

By the same way, $g^k = \boldsymbol{I} + [\vec{0} \cdots \vec{0}\ \boldsymbol{m}^k\ \vec{0} \cdots \vec{0}] = \boldsymbol{I} + \boldsymbol{M}^k$ because $\boldsymbol{z}^{\mathsf{T}}(g^k - \boldsymbol{I})$ is a sparse vector. $\square$

**Lemma B.15.** (*W 2 & MA 2*) *Let $g^k = I + M^k \in GL(n)$, where only the $k$-th column of $M^k$ can be nonzero. If $g^k$ has finite order, then either $g^k = I$ or $(g^k)^2 = I$. Therefore, $g^k$ cannot have order greater than two.*

*Proof.* (**W 2** & **MA 2**) Since only one column of $M^k$ can be nonzero, $\mathrm{rank}(g^k - I) \leq 1$. Because $g^k$ has finite order, it is diagonalizable over $\mathbb{C}$ and all its eigenvalues are roots of unity. The rank condition implies that at most one eigenvalue of $g^k$ differs from 1 (i.e., its eigenvalues are $1, \ldots, 1, \lambda_*$, where $\lambda_*$ is the only eigenvalue that may differ from 1). Since $g^k$ is real, every non-real eigenvalue must occur together with its complex conjugate. Therefore, the only possible eigenvalue different from 1 is $-1$.

If all eigenvalues are 1, diagonalizability implies $g^k = I$. Otherwise, the eigenvalues belong to $\{1, -1\}$, which implies $(g^k)^2 = I$. Hence, every non-identity finite-order element of this form has order two. $\square$

Since an injective group homomorphism preserves the order of a group element, the image of a generator of $G^{F^i}$ must have order $|G^{F^i}| > 2$. This contradicts Lemma B.15.

**Theorem B.16.** (*Limit of $GL'(n)$*) *According to Proposition Theorem B.11, if $g \in GL'(n)$ is compatible with Conditions 4.1–4.3 and, $|G^{F^i}| > 2$, then $g = I$.*

*Proof.* Through the Lemma Theorem B.12 to Lemma Theorem B.15, if the group element $g$ represents the element of cyclic group, then only the identity matrix satisfies the Eq. Equation (10). There is always a case as $g = e^{\mathfrak{g}}$, and $(g)^{n-1} = (e^{\mathfrak{g}})^{n-1}$ but $(g)^n \neq (e^{\mathfrak{g}})^n$, where $g \neq I$, because $g$ can not represent the element of cyclic group. By the equivariant function $q_\phi$:

$$
\begin{aligned}
q_\phi(x_1) &= z_1 \\
q_\phi(g_x \circ_x x_1) &= \Gamma(g_x) \circ_z z_1 \\
&\vdots \\
q_\phi(g_x^{n-1} \circ_x x_1) &= [\Gamma(g_x)]^{n-1} \circ_c z_1 \\
q_\phi(g_x^n \circ_x x_1) &\neq [\Gamma(g_x)]^n \circ_c z_1 \ (\because [\Gamma(g_x)]^n \neq (e^{\mathfrak{g}})^n).
\end{aligned}
\tag{23}
$$

It implies that element of cyclic group $g_x$ between the $x_k$ and $x_{k+1}$ is divided as:

$$
\Gamma(g_x) = \begin{cases} e^{\mathfrak{g}} & \text{if } k < n \\ e^{\mathfrak{g}'} & \text{if } k = n \end{cases} \quad , \text{ where } \mathfrak{g} \neq \mathfrak{g}'.
\tag{24}
$$

$\square$

Therefore, representing the cyclic group of the dataset with representational consistency of identical transformations is impossible according to the Theorem B.16.

**Limitations of $GL(n)$ to Represent the Cyclic Group on the Disentangled Space**

**Theorem B.17.** (*Limitation of $GL(n)$*) *If Conditions 4.1–4.3 hold, $|G^{F^i}| > 2$, and $H = \{h \in GL(n) \mid h = I + M^i\}$, then $h = I$ is the only element of $H$.*

*Proof.* If the invertible matrix $g$ changes a single dimension value of the latent vector $\boldsymbol{z}$ then,

$$
\begin{aligned}
\boldsymbol{z}^\mathsf{T} g - \boldsymbol{z} &= [0, \ldots, 0, \alpha, 0, \ldots, 0] \\
\Rightarrow \boldsymbol{z}^\mathsf{T}(g - I) &= [0, \ldots, 0, \alpha, 0, \ldots, 0]. \\
\Rightarrow [z_1, \ldots, z_n] \begin{bmatrix} g_{11} - 1 & \cdots & g_{1n} \\ \vdots & \ddots & \vdots \\ g_{n1} & \cdots & g_{nn} - 1 \end{bmatrix} &= [0, \ldots, 0, \alpha, 0, \ldots, 0]
\end{aligned}
\tag{25}
$$

As defined the $G$-set as a latent vector space $\mathcal{Z}$, group element $g$ satisfies the Eq. Equation (25) over all vectors. Then $g - \boldsymbol{I}$ elements are all 0, except the $k^{th}$ column vector ($\boldsymbol{m}^k \neq 0$). Therefore, group element $g \in H$. $\qquad\square$

**Theorem B.18.** *If $h \in H \backslash \{\boldsymbol{I}\}$, then this set does not represent the representational consistency of identical transformations with element of cyclic group.*

*Proof.* According to Lemma Theorem B.15, $h^k = \boldsymbol{I}$. Therefore, $h^k$ represents only the representational consistency of identical transformations of the dataset space. $\qquad\square$

Therefore, representing the cyclic group of the dataset with representational consistency of identical transformations implemented by the $GL(n)$ is impossible except in the case of the identity matrix.

## B.3 Proof: Limitations of Vector Addition for Cyclic Group on the Disentangled Space

In the previous works (Hwang et al., 2023), the vector addition is used to define a group action between two latent vectors for an equivariant model, where the group action $act(g, \boldsymbol{z}) = g + \boldsymbol{z}$.

**Theorem B.19.** *If the group action is defined as $act(g, \boldsymbol{z}_i) = g + \boldsymbol{z}_i$, then $g$ does not represent the representational consistency of identical transformations for element of cyclic group with disentangled representation, where $g \in G \backslash \{\vec{0}\}$ and $\boldsymbol{z} \in \mathbb{R}^n$.*

*Proof.* If $g$ represents the element of cyclic group of $\mathcal{X}$, then there exists:

$$(g)^n = ng = \vec{0}. \tag{26}$$

The solution of Eq. Equation (26) is $g = \vec{0}$. There is always a case as $g = \boldsymbol{a}$, and $(g)^{n-1} = (n-1)\boldsymbol{a}$ but $(g)^n \neq n\boldsymbol{a}$, where $g \neq \vec{0}$, because $g$ can not represent the element of cyclic group. By the equivariant function $q_\phi$:

$$
\begin{aligned}
q_\phi(x_1) &= z_1 \\
q_\phi(g_x \circ_x x_1) &= \Gamma(g_x) \circ_z z_1 \\
&\vdots \\
q_\phi(g_x^{n-1} \circ_x x_1) &= (n-1)[\Gamma(g_x)] \circ_c z_1 \\
q_\phi(g_x^n \circ_x x_1) &\neq n[\Gamma(g_x)] \circ_c z_1 \ (\because n[\Gamma(g_x)] \neq n\boldsymbol{a}).
\end{aligned}
\tag{27}
$$

It implies that element of cyclic group $g_x$ between the $x_k$ and $x_{k+1}$ is divided as:

$$\Gamma(g_x) = \begin{cases} \boldsymbol{a} \text{ if } k < n \\ \boldsymbol{a}' \text{ if } k = n \end{cases} , \text{ where } \boldsymbol{a} \neq \boldsymbol{a}'. \tag{28}$$

$\qquad\square$

According to Theorem B.19, the group action defined as $act(g, \boldsymbol{z}) = g + \boldsymbol{z}$ represents the cyclic group of the dataset with representational consistency of identical transformations when $g = \vec{0}$. However, the group elements are insufficient to encompass the entire cyclic group of input space. Additionally, this causes the inconsistency issue when holding consistency with vector addition.

## B.4 Proof: Loss of Factor-State Distinguishability under Non-Injective Mappings

**Proposition B.20.** *Let $b : \mathcal{Z} \to \mathcal{Y}$ be surjective but non-injective. Suppose that there exist two factor states $fv_i$ and $fv_j$ that differ in the $k$-th factor and satisfy $b(h_\phi(fv_i)) = b(h_\phi(fv_j))$. Then $b \circ h_\phi$ cannot distinguish these two values of the $k$-th factor and therefore does not provide a faithful dimension-wise representation of that factor.*

*Proof.* The two factor states have different values of the $k$-th factor but are mapped to the same representation in $\mathcal{Y}$. Therefore, no function of $b \circ h_\phi$ can distinguish their $k$-th factor values. Hence, the $k$-th factor is not faithfully represented. $\square$

## C  Details of the Proposed Method Motivation

### C.1  Why a Bijective Cyclic Representation?

**From Factor States to Cyclic Group Structure.**  As established in Section 4, the symmetry group acting on the factor states is $G^F \cong \mathbb{Z}_{|F^1|} \times \cdots \times \mathbb{Z}_{|F^n|}$. A cyclic group $G = \{e, g_1, g_2, \ldots, g_{n-1}\}$ has the convenient property that all elements are integer powers of a single generator; consequently, if the model learns one generator, it implicitly represents the entire group. Motivated by Yang et al. (2021), we implement the cyclic group as the $N$-th roots of unity. Concretely, the cyclic group is

$$G^c = G_1^c \times G_2^c \times \cdots \times G_k^c, \quad G_i^c = \left\{ g_i^c \mid g_i^c = \tfrac{2\pi}{N} k', \ k' \in \{0, 1, \ldots, N-1\} \right\}, \tag{29}$$

where $N \in \mathbb{Z}^+$. The group action is defined as $act : \Theta^n \times G^c \to \Theta^n$, $act(g^c, \boldsymbol{\theta}) = g^c + \boldsymbol{\theta}$, where $\boldsymbol{\theta} \in \Theta^n$ and $-\pi < \boldsymbol{\theta}^i \leq \pi$.

**Isomorphism, not Homomorphism.**  The surjective-mapping family (Case 3 in Section 4.2) represents cyclic symmetries via a homomorphism $G^z \to G^y$, which can collapse distinct elements and lose group structure. To avoid this, we require an isomorphism $G^z \cong G^F$, i.e., a bijective structure-preserving map between the latent group and the factor group. This isomorphism requirement directly motivates working on the unit circle $S^1$ via a bijective (rather than surjective) embedding.

**Connecting to Definition 3.1.**  The key property of the bijective cyclic representation is that identical transformations map to identical group elements regardless of the starting state. We formalize this as the following proposition.

**Proposition C.1** (Consistent Representation via Bijective Cyclic Embedding). *Let $f' \circ f : \mathbb{R} \to \Theta$ be the bijective map defined in Section 5.2 (Cayley transform composed with angle extraction), and let $\boldsymbol{\theta}'^j = \arg\min_{\boldsymbol{V}^i \in \boldsymbol{V}} |\boldsymbol{V}^i - \boldsymbol{\theta}^j|$ be the nearest-neighbor projection onto the fixed grid $V$ defined in Section 5.3. Then, under the cyclic group structure $G^{F^i} \cong \mathbb{Z}_N$, the resulting latent representation satisfies Definition 3.1: for any two pairs $(f_{v_i}, f_{v_j})$ and $(f_{v_n}, f_{v_m})$ with $f_{v_j} = g^F \cdot f_{v_i}$ and $f_{v_m} = g^F \cdot f_{v_n}$, the corresponding latent transformation $\Gamma(g^F) = \tfrac{2\pi}{N} k'$ depends only on $g^F$ (equivalently, on the step $k$), and not on the particular pair.*

*Proof sketch.* Because $f' \circ f$ is bijective, each real-valued latent $z$ maps to a unique angle $\boldsymbol{\theta} \in \Theta^n$. The fixed grid $\boldsymbol{V}$ partitions $\Theta$ into $N$ equal-width bins of size $\tfrac{2\pi}{N}$, so any angle is snapped to one of the $N$ canonical grid points. The group action $act(g^c, \boldsymbol{\theta}) = g^c + \boldsymbol{\theta}$ with $g_i^c = \tfrac{2\pi}{N} k'$ then advances the grid point by exactly $k'$ steps. Since the grid is fixed (not sample-dependent), the same $k'$-step transformation always produces the same latent shift $\tfrac{2\pi}{N} k'$, independently of the starting angle $\boldsymbol{\theta}_i$ or $\boldsymbol{\theta}_n$. This directly satisfies the pair-independence condition in Definition 3.1. $\square$

## D  Details of Experiments Setting

### D.1  Resources

We set the following settings for all experiments on a single Galaxy 2080Ti GPU, a single Galaxy 3090 GPU, and a single NVIDIA A100 GPU for the dSprites 3D Shapes and MPI3D datasets. The Python version is 3.7.10, and the PyTorch version is 1.9.1. More details are in the README.md file.

### D.2  Datasets

1) The dSprites dataset consists of 737,280 binary $64 \times 64$ images with five independent ground truth factors(number of values), *i.e.* x-position (32), y-position (32), orientation (40), shape (3), and scale (6) (Matthey

et al., 2017). Any composite transformation of x- and y-position, orientation (2D rotation), scale, and shape is commutative. 2) The 3D Shapes dataset consists of 480,000 RGB $64 \times 64 \times 3$ images with six independent ground truth factors: orientation (15), shape (4), floor color (10), scale (8), object color (10), and wall color (10) (Burgess & Kim, 2018). 3) The MPI3D (real-world complex) dataset consists of 460,800 RGB $64 \times 64 \times 3$ images with seven independent ground truth factors: color (4) shape (4), size (2), height (3), background color (3), horizontal (40), and vertical axis (40) (Gondal et al., 2019). Additionally, we use cLPR dataset[1] consists of 250,047 RGB $64 \times 64 \times 3$ images with three independent ground truth factors: x-rotation (63), y-rotation (63), and z-rotation (63).

### D.3 Setting for compositional Generalization

**Train and Test datasets**  We except the case [shape=$ellips$, position-x $\geq 0.6$, position-y $\geq 0.6, 120° \leq$ rotation $\leq 240°$, scale $< 0.6$] for dSprites r2e training set and [shape=$square$, position-x $\geq 0.5$] for dSprites r2r training set.

We except the case [floor-hue $>$ 0.5, wall-hue $>$ 0.5, object-hue $\geq$ 0.5, shape=cylinder, object-scale=1, object-orientation=0] for 3D Shapes r2e training set and [object-hue $\geq 0.5$, shape=oblong] for 3D Shapes r2r training set.

We except the case [shape $=$ cone, object size $=$ 0, cameraheight $=$ 1, background color $=$ purple, object color $\in$ {blue, brown, olive}, horizontal axis $\geq$ 20, vertical axis $\geq$ 20]for r2e training set and [shape $=$ cylinder, scale $< 6$, orientation, $16 \leq$ horizontal axis $< 32$, vertical axis] for r2r training set.

**Hyper-Parameter Tuning**  We set $\beta \in \{1, 2, 10\}$ for $\beta$-VAE (Higgins et al., 2017) and $\beta$-TCVAE Chen et al. (2018), and $\beta \in \{1, 10\}$ for VAE-MAGA, which employs the MAGA-net proposed module on the CNN-based encoder and decoder, instead of Glow (Kingma & Dhariwal, 2018). We set the common hyper-parameters of the proposed method at $\alpha \in \{100, 1000\}$, $\gamma \in \{1, 10\}$, $\lambda \in \{0.0, 1.0\}$ for the supervised and ground truth models, and $\beta \in \{1.0, 2.0\}$ for the supervised method. We run each model with three seeds, $\in \{1, 2, 3\}$. We set $N = 1000$ and $n' = 10$.

**Decoder Equivariant Loss**  For compositional generalization, we add the decoder equivariant loss as:

$$\mathcal{L}_{de} = R.E(x_j, p_\theta \circ g_{i \to j} \circ_\theta (f' \circ f \circ q_\phi(x_i))), \tag{30}$$

where $R.E(\cdot)$ is a reconstruction error and $p_\theta$ is a decoder. We add the $\mathcal{L}_{de}$ to the objective losses with hyper-parameter $\lambda$.

### D.4 Setting for Disentanglement Learning

**Hyper-Parameter Tuning**  We set $\beta \in \{1, 2, 10\}$ for $\beta$-VAE (Higgins et al., 2017) and $\beta$-TCVAE (Chen et al., 2018), $\gamma \in \{10, 20, 40\}$ for Factor-VAE Kim & Mnih (2018), hy$_{\text{rec}} \in \{0.1, 0.2, 0.7\}$ for CLG-VAE (Zhu et al., 2021), $\beta = 1$ for Ada-GVAE (Locatello et al., 2020a). We set the common hyper-parameters of proposed method $\alpha \in \{100, 1000\}$, $\gamma = 1$ for superivsed and ground truth model, $\beta \in \{1.0, 2.0\}$ for supervised method, and $\lambda = 1.0, p = 0.5$ for weakly-supervised method. We run 10 seed variance over each model with seed $\in \{1, 2, \ldots, 10\}$. We set $N = 1000$ and $n' = 10$. We evaluate four metrics $\beta$-VAE metric (Higgins et al., 2017), Factor VAE metric (Kim & Mnih, 2018), SAP (Kumar et al., 2018), and DCI (Eastwood & Williams, 2018).

## E compositional Generalization Results

As shown in Figure 9a, we selected the four worst samples (those with the highest reconstruction errors): 1) $\beta$-VAE results contain only position semantics, 2) $\beta$-TCVAE captures the position and scale values but fails to capture the shape and rotation factors, 3) VAE-MAGA struggles with generalization. Even though our method does not capture all semantics, it shows improvement compared to the baselines: 4) the supervised method misses either the shape or rotation, and 5) the GT model only misses the shape semantic. As shown

---

[1]https://github.com/yvan/cLPR

in Figure 9b, the representations of the baseline are not close to a disentangled representation. In contrast, the representation of the supervised method approaches a disentangled representation and shows better generalization. This implies that a disentangled representation containing the symmetry structure could benefit compositional generalization.

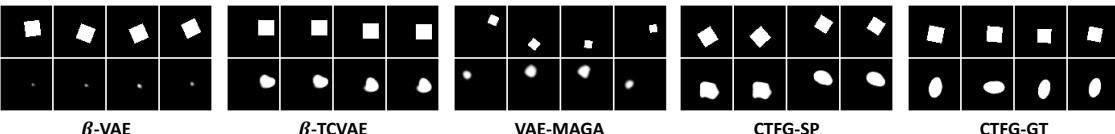

(a) Visualization of generated images of the worst 4 samples of R2R task. Each $1^{st}$ row is a group truth, and $2^{nd}$ row is a generated image. The red box indicates the results, which do not contain all the semantics of ground truth.

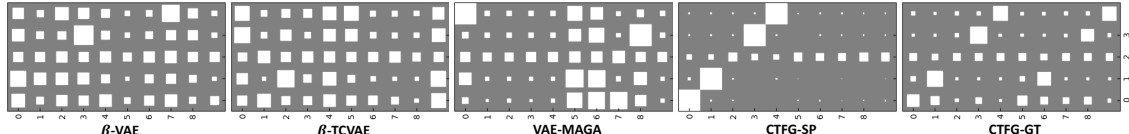

(b) Visualization of DCI metric of dSprites dataset

Figure 9: Qualitative results of compositional generalization of dSprites dataset (R2R). A more sparse matrix implies clear disentanglement.

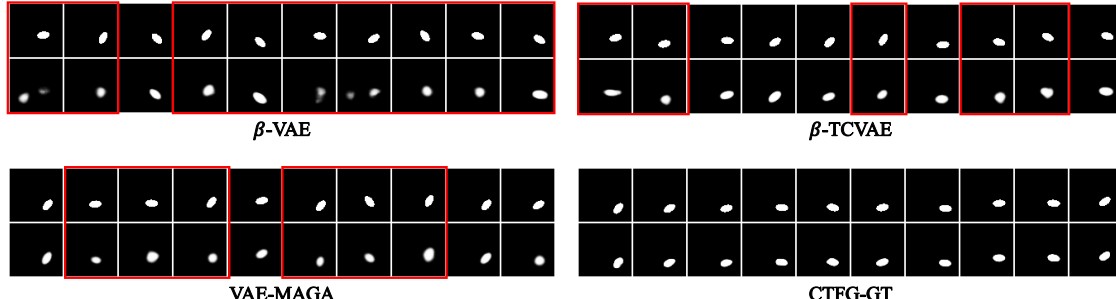

(a) Visualization of generated images of the worst 4 samples of R2E task. Each $1^{st}$ row is a group truth, and $2^{nd}$ row is a generated image. The red box indicates the results, which do not contain all the semantics of ground truth.

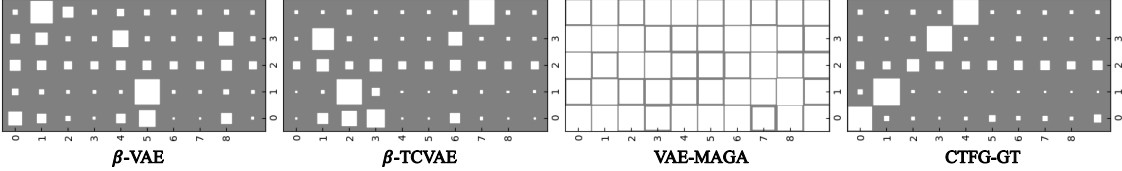

(b) Visualization of DCI metric of dSprites dataset

Figure 10: Qualitative results of compositional generalization of dSprites dataset (R2E). A more sparse matrix implies clear disentanglement.

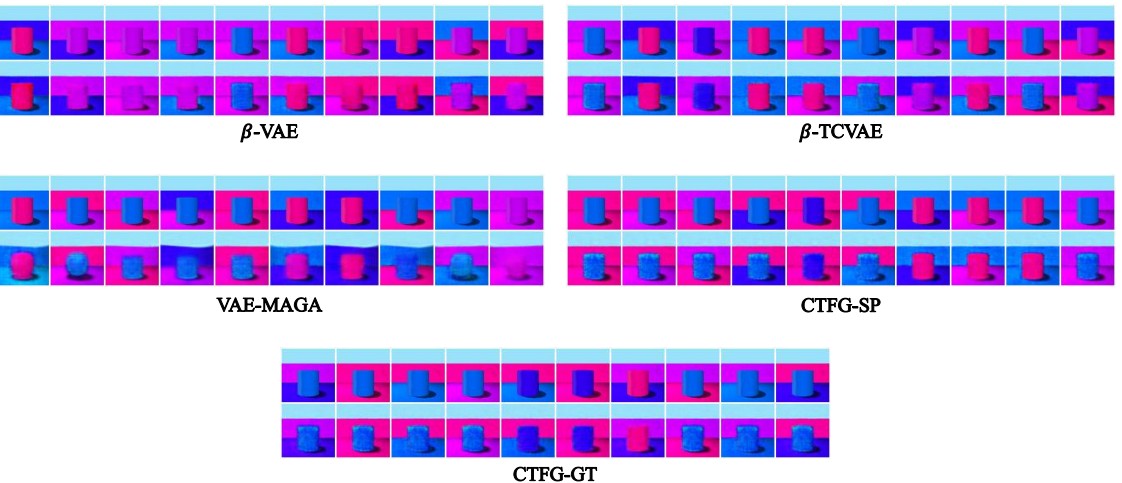

(a) Visualization of generated images of the worst 10 samples of the R2R task. Each $1^{st}$ and $2^{nd}$ row shows the group truth samples and the generated results, respectively. The red box indicates the negative results, which do not contain all the semantics of the ground truth. We utilize randomly selected pivot images as introduced in Hwang et al. (2023) for the CTFG model.

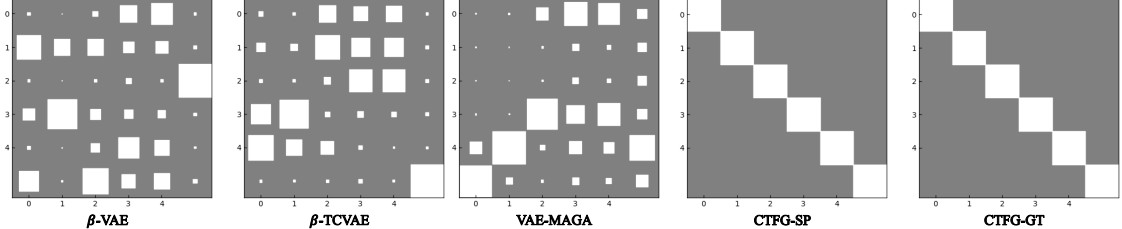

(b) Visualization of DCI metric of 3D Shapes dataset (training set). A more sparse matrix implies clear disentanglement.

Figure 11: Qualitative results of compositional generalization of 3D Shapes dataset (R2E).

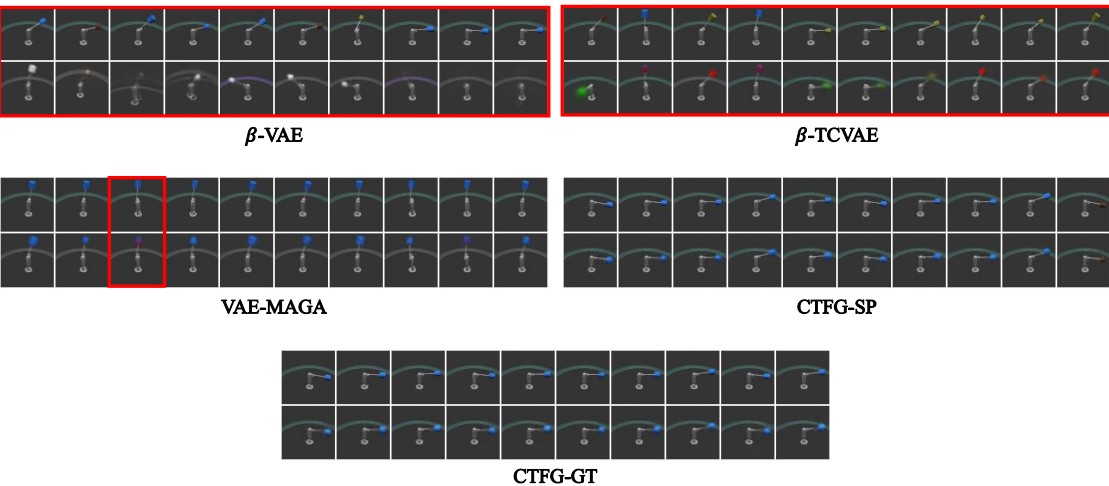

(a) Visualization of generated images of the worst 10 samples of the R2R task. Each $1^{st}$ and $2^{nd}$ row shows the group truth samples and the generated results, respectively. The red box indicates the negative results, which do not contain all the semantics of the ground truth. We utilize randomly selected pivot images as introduced in Hwang et al. (2023) for the CTFG model.



(b) Visualization of DCI metric of 3D Shapes dataset. A more sparse matrix implies clear disentanglement.

Figure 12: Qualitative results of compositional generalization of MPI3D dataset (R2E).

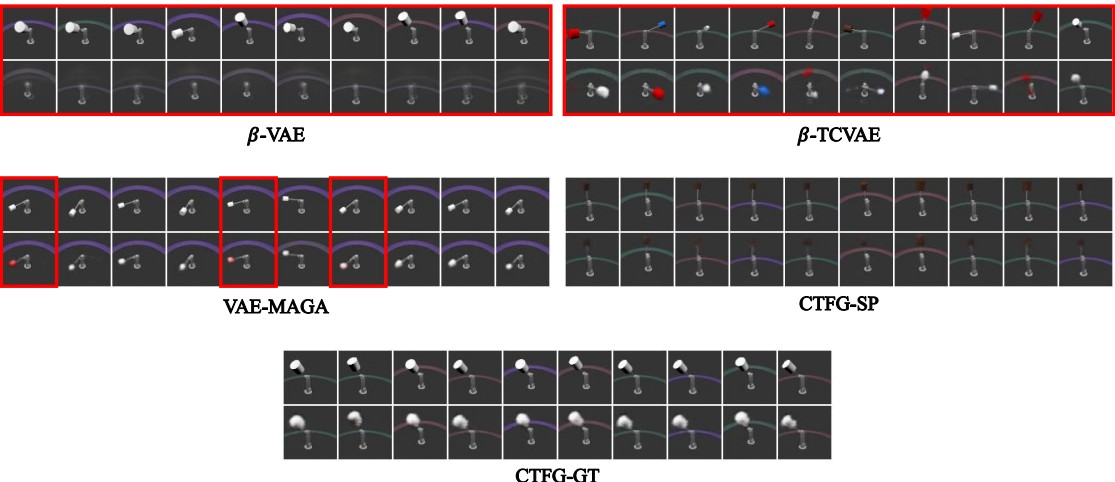

(a) Visualization of generated images of the worst 10 samples of the R2R task. Each $1^{st}$ and $2^{nd}$ row shows the group truth samples and the generated results, respectively. The red box indicates the negative results, which do not contain all the semantics of the ground truth. We utilize randomly selected pivot images as introduced in Hwang et al. (2023) for the CTFG model.



(b) Visualization of DCI metric of 3D Shapes dataset. A more sparse matrix implies clear disentanglement.

Figure 13: Qualitative results of compositional generalization of MPI3D dataset (R2R).

## F  Disentanglement Learning Results

### F.1  Trade-Off (3D Shapes)

As illustrated in Figure 14, the proposed models improve the reconstruction error and disentanglement performance simultaneously on the dSprites dataset. Additionally, while the reconstruction error slightly decreases, the model performance dramatically improves compared to the baselines on the 3D Shapes dataset.

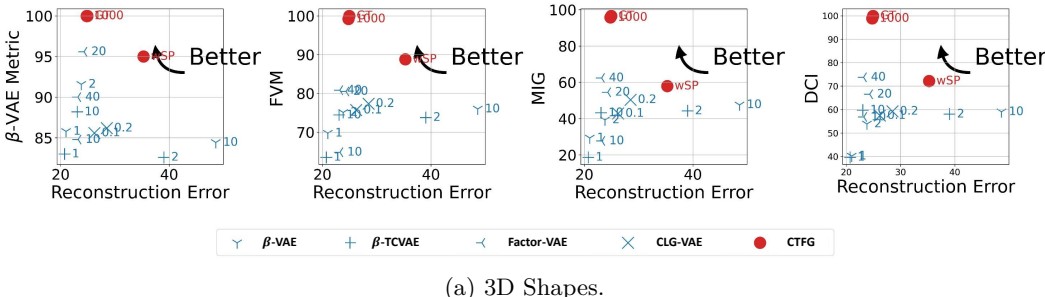

(a) 3D Shapes.

Figure 14: Reconstruction error vs. evaluation metrics ($\beta$-VAE metric, FVM, MIG, and DCI). The top left side indicates the best results for both objectives.

Table 9: Disentanglement performance of Homomorphism VAE vs. Ours with dSprites.

| Model | beta-VAE | FVM | MIG | DCI |
|---|---|---|---|---|
| Homomorphism VAE | 18.80($\pm$5.75) | 30.24($\pm$12.18) | 0.39($\pm$0.76) | 1.35($\pm$1.12) |
| Groupified-VAE | 79.30($\pm$9.23) | 69.75($\pm$13.66) | 21.03($\pm$9.20) | 31.08($\pm$10.87) |
| CTFG-SP | **91.40**($\pm$4.99) | **93.74**($\pm$1.82) | **51.02**($\pm$2.42) | **64.69**($\pm$1.55) |

### F.2  Ours vs. Homomorphism VAE

As shown in Table Table 9, our model outperforms Homomorphism VAE (Keurti et al., 2023) and Groupified-VAE (Yang et al., 2021). Homomorphism VAE (Keurti et al., 2023) utilizes s the SO(2) for disentangled representation, and the elements of SO(2) affect the multi-dimension value of the latent vector. It implies that the SO(2) group is not appropriate symmetry for dimension-wise disentangled representation.

### F.3  Impact of Known Label Ratio

We set the known label ratio $p \in \{0.1, 0.2, 0.4, 0.5\}$. As shown in Table 10, our model is robust to the known label ratio except for the DCI metric.

Table 10: Disentanglement performance of weakly-supervised learning methods.

| Model | dSprites | | | | 3DShapes | | | |
|---|---|---|---|---|---|---|---|---|
| | beta-VAE | FVM | MIG | DCI | beta-VAE | FVM | MIG | DCI |
| Ada-GVAE | 83.60($\pm$2.61) | 83.67($\pm$2.97) | 21.34($\pm$5.35) | **47.26**($\pm$1.89) | 72.75($\pm$6.50) | 59.81($\pm$6.14) | 24.77($\pm$7.48) | 64.57($\pm$4.04) |
| CTFG-wSP (0.1) | **88.60**($\pm$7.72) | 83.36($\pm$3.51) | 14.71($\pm$1.25) | 23.46($\pm$1.69) | **86.80**($\pm$3.90) | **84.05**($\pm$2.66) | **55.17**($\pm$2.18) | 61.79($\pm$3.15) |
| CTFG-wSP (0.2) | **89.78**($\pm$6.67) | **83.88**($\pm$2.76) | **23.03**($\pm$2.54) | 28.07($\pm$1.40) | **85.00**($\pm$13.11) | 83.00($\pm$7.29) | 54.20($\pm$13.35) | 60.26($\pm$12.00) |
| CTFG-wSP (0.4) | **88.20**($\pm$5.92) | 83.49($\pm$2.22) | **31.87**($\pm$2.02) | 39.44($\pm$0.79) | **86.40**($\pm$6.98) | **87.61**($\pm$7.09) | **61.47**($\pm$9.51) | **67.87**($\pm$8.39) |
| CTFG-wSP (0.5) | **87.00**($\pm$7.07) | **84.50**($\pm$1.41) | **31.95**($\pm$2.40) | 39.36($\pm$1.49) | **95.00**($\pm$7.07) | **88.81**($\pm$13.17) | 57.94($\pm$16.52) | **72.14**($\pm$3.23) |

### F.4  Impact of Each Loss

## G  Qualitative Analysis of Disentanglement Learning

**3D Shapes**  As shown in Figure 15, the baseline results show that multiple factors are changed when a single dimension value is changed. On the other hand, ours show the fully disentangled results represent: $1^{st}$ row is the floor color changes, $2^{nd}$ row is the wall color changes, $3^{rd}$ row is the object color changes, $4^{th}$ row is the scale of object, $5^{th}$ row is the object shape changes, and $6^{th}$ row is the rotation changes.

Table 11: Disentanglement performance over hyper-parameters

| $(\alpha, \gamma)$ | reconst. err. | beta-VAE | FVM | MIG | DCI |
|---|---|---|---|---|---|
| (100, 1) | **17.88**($\pm$1.24) | 88.40($\pm$4.97) | 82.13($\pm$1.86) | 33.88($\pm$3.03) | 42.27($\pm$2.02) |
| (200, 1) | 21.23($\pm$0.89) | 92.44($\pm$5.81) | 87.54($\pm$4.13) | 35.21($\pm$2.17) | 47.40($\pm$2.14) |
| (500, 1) | 26.00($\pm$2.72) | 94.00($\pm$4.42) | 96.41($\pm$1.83) | 43.73($\pm$3.55) | 55.36($\pm$2.70) |
| (1000, 1) | 27.41($\pm$0.73) | **95.80**($\pm$4.57) | **99.26**($\pm$1.12) | **51.81**($\pm$2.97) | **63.26**($\pm$2.73) |

(a) dSrites with Ground Truth model

| $(\alpha, \beta, \gamma)$ | reconst. err. | beta-VAE | FVM | MIG | DCI |
|---|---|---|---|---|---|
| (100, 1, 1) | **19.53**($\pm$1.75) | 85.78($\pm$5.87) | 82.10($\pm$2.81) | 27.12($\pm$1.98) | 34.42($\pm$1.04) |
| (100, 2, 1) | 17.58($\pm$1.08) | 86.44($\pm$5.90) | 82.40($\pm$1.62) | 34.08($\pm$1.75) | 41.16($\pm$0.98) |
| (1000, 1, 1) | 26.22($\pm$1.31) | **91.40**($\pm$4.90) | 93.46($\pm$1.97) | 46.35($\pm$1.94) | 57.92($\pm$1.97) |
| (1000, 2, 1) | 26.32($\pm$2.20) | **91.40**($\pm$4.99) | **93.74**($\pm$1.82) | **51.03**($\pm$2.42) | **64.69**($\pm$1.55) |

(b) dSrites with Supervised method

| $(\alpha, \gamma)$ | reconst. err. | beta-VAE | FVM | MIG | DCI |
|---|---|---|---|---|---|
| (100, 1) | 33.62($\pm$5.38) | 89.60($\pm$6.17) | 82.44($\pm$3.78) | 53.37($\pm$12.87) | 60.04($\pm$13.98) |
| (200, 1) | 34.50($\pm$5.35) | 95.00($\pm$5.34) | 91.95($\pm$7.14) | 68.06($\pm$17.82) | 74.16($\pm$14.69) |
| (500, 1) | 29.30($\pm$1.72) | **100.00**($\pm$0.00) | 97.28($\pm$3.05) | 86.68($\pm$5.56) | 90.68($\pm$5.87) |
| (1000, 1) | **24.94**($\pm$1.51) | **100.00**($\pm$0.00) | **100.00**($\pm$0.00) | **95.57**($\pm$0.80) | **99.94**($\pm$0.18) |

(c) 3D Shapes with Ground Truth model

**MPI3D**   As shown in Figure 16, the baseline results show that multiple factors are changed when a single dimension value is changed. Also, the object usually disappeared following the intervals with baselines. On the other hand, supervised methods show better results than baselines: $1^{st}$ row is the object color changes, $2^{nd}$ row is the object shape changes, $3^{rd}$ row is the object size changes, $5^{th}$ row is the background color changes, $7^{th}$ row is the vertical axis changes, and $9^{th}$ row is the height changes. Also, the GT model results represent: $1^{st}$ row is the object color changes, $2^{nd}$ row is the object shape changes, $3^{rd}$ row is the object size changes, $4^{th}$ row is the height changes, $5^{th}$ row is the background color changes, $6^{th}$ row is the vertical axis changes, and $7^{th}$ row is the horizontal axis changes.

**cLPR**   As shown in Figure 17, the Homomorphism VAE (Keurti et al., 2023) shows that multiple factors are changed when a single dimension value is changed. Also, the reconstruction quality is lower than ours (CTFG-GT and CTFG-SP). On the other hand, the supervised method shows better results than the Homomorphism VAE: $1^{st}$ and $3^{rd}$ rows are z-axis rotation, $2^{nd}$ row is y-zis rotation, and $4^{th}$ row is x-axis rotation. Also, the GT model results represent: $1^{st}$ and $2^{nd}$ rows are x-axis rotation, $3^{rd}$ row is z-axis rotation, and $4^{th}$ and $6^{th}$ rows are y-axis rotation.

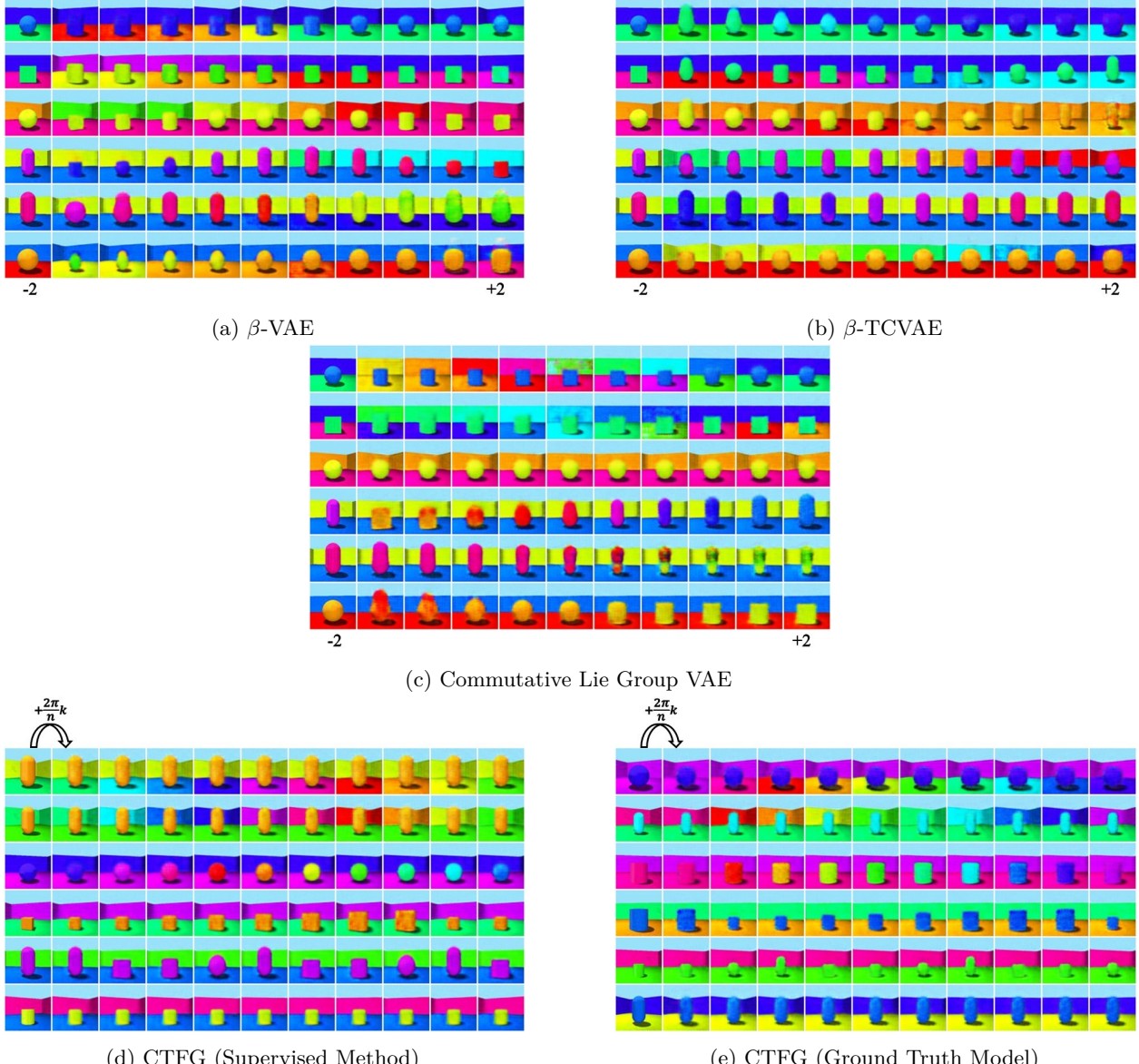

(a) $\beta$-VAE

(b) $\beta$-TCVAE

(c) Commutative Lie Group VAE

(d) CTFG (Supervised Method)

(e) CTFG (Ground Truth Model)

Figure 15: The $1^{st}$ column images are randomly selected from the dataset. Each row indicates each dimension of each model. $\beta$-VAE, $\beta$-TCVAE, and Commutative Lie Group VAE trace each dimension value from -2 to +2. The proposed methods apply a group action $+\frac{2\pi}{n}$ to the selected images a total of 10 times.

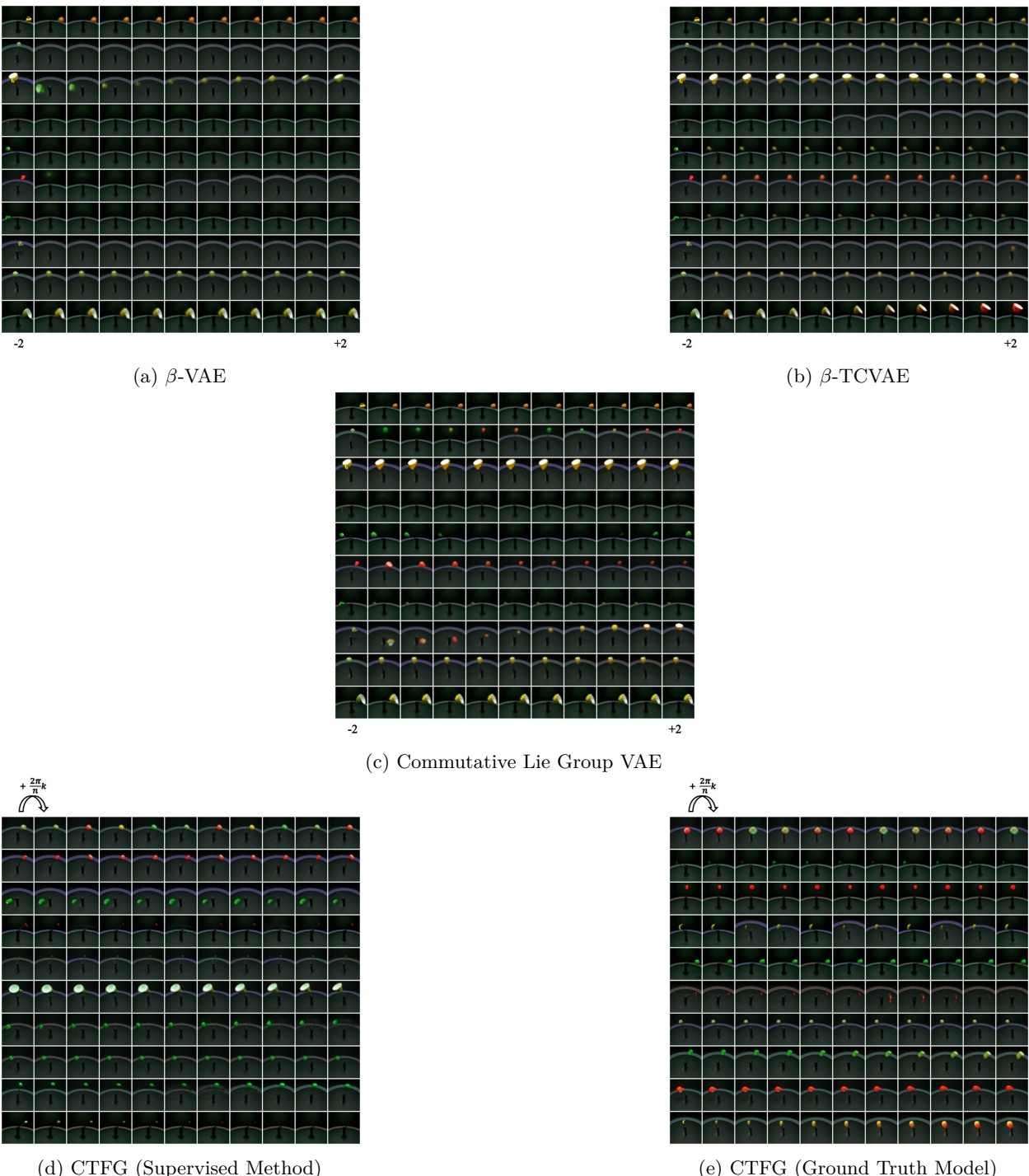

(a) $\beta$-VAE

(b) $\beta$-TCVAE

(c) Commutative Lie Group VAE

(d) CTFG (Supervised Method)

(e) CTFG (Ground Truth Model)

Figure 16: The $1^{st}$ column images are randomly selected from the dataset. Each row indicates each dimension of each model. $\beta$-VAE, $\beta$-TCVAE, and Commutative Lie Group VAE trace each dimension value from -2 to +2. The proposed methods apply a group action $+\frac{2\pi}{n}$ to the selected images a total of 10 times.

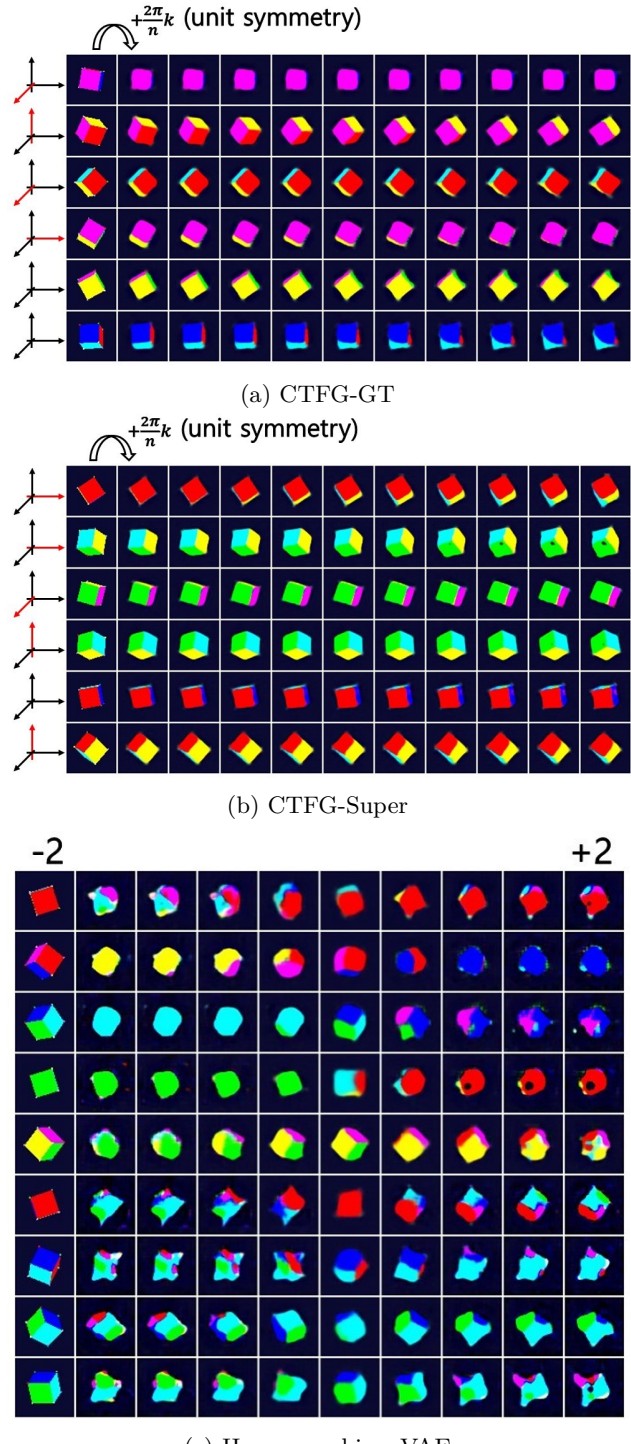

(a) CTFG-GT

(b) CTFG-Super

(c) Homomorphism VAE

Figure 17: The $1^{st}$ column images are randomly selected from the dataset. Each row indicates each dimension of each model. CTFG-GT, CTFG-Super ($\alpha$: 100.0), and homomorphism VAE trace each dimension value from -2 to +2. The proposed methods apply a group action $+\frac{2\pi}{n}$ to the selected images a total of 10 times. And red color axis is a rotation axis.

