# OpenReview forum: "The Impact of Enforcing Representational Consistency of Identical Transformations for Disentangled Representation"
_TMLR — Decision pending for TMLR_

### Review · Reviewer_FwEB · 2026-05-08

**Summary Of Contributions:**

This paper identifies a critical issue in symmetry-based disentanglement learning: representational inconsistency of identical transformations. The authors formalize this issue using a group-theoretic framework and demonstrate that in existing methods, the same semantic change(e.g., rotation) is represented differently across different sample pairs.

The key contributions include:

ⅠTheoretical Analysis: Proving the limitations of three common symmetry parameterizations（GL(n), vector-additive, and surjective mappings）in maintaining consistency for cyclic groups.

ⅡProposed Framework(CTFG): Introducing a bijective mapping to the complex unit circle via the Cayley transform and a fixed-grid codebook to enforce discrete, consistent transformations.

Ⅲ Empirical Validation: Demonstrating near-perfect symmetry consistency and significantly improved compositional generalization on benchmarks like dSprites, 3D Shapes, and MPI3D.
The paper further claims that enforcing representational consistency improves both disentanglement and compositional generalization performance.


Strengths:

1.	The paper introduces a clear formalization of the “representational consistency of identical transformations”.

2.	It provides a unified analysis of multiple symmetry parameterization families(Section 4).

3.	The empirical evaluation is extensive across multiple datasets and tasks.

Weaknesses:

1.	The proposed method is composed of multiple strong inductive biases, making it difficult to isolate the contribution of the proposed “consistency principle” (Section 5.2–5.5).

2.	The theoretical analysis of prior methods is not fully rigorous and appears to rely on strong assumptions without complete formal justification.

3.	The focus is primarily on cyclic groups, which may not cover all real-world variations.

**Additional Comments:**

Confidence: Low

**Audience:**

Yes

**Audience Explanation:**

The paper addresses core topics in the TMLR community, including symmetry-based representation learning, disentanglement, and compositional generalization. Despite limitations in theoretical rigor and causal validation, the proposed perspective on transformation consistency may still be of interest. The novel technical approach will interest researches working on VAEs, group theory in ML, and robust representation learning.

**Broader Impact Concerns:**

There are no major ethical concerns. The paper focuses on fundamental representation learning on standard benchmark datasets.

**Claims And Evidence:**

No

**Claims Explanation:**

While the experimental results show strong performance, the evidence does not convincingly demonstrate that the observed improvements are a causal result of the proposed “consistency principle” being learned.

The near-perfect consistency appears to be a direct mathematical consequence of the architectural constraints and supervision rather than an emergent property. The metrics is likely influenced by Ground-truth-based step supervision, Fixed-grid quantization and Canonical anchor loss. However, the paper does not clearly isolate the contribution of each component, making it difficult to attribute improvements specifically to the proposed “consistency principle”.

**Requested Changes:**

Major Concerns:

1.	Clarify whether consistency is learned or enforced. The current formulation does not clearly distinguish whether representational consistency emerges from learning or is directly imposed by the training design. Add ablation studies and demonstrate whether consistency still holds without these constraints.

2.	Strengthen theoretical claims in Section 4.

Minor Concerns:

1.	Clarify the choice of L1 vs L2 loss in Equations 9 and 10.

2.	The fixed-grid quantization may significantly influence performance independently of symmetry modeling. It is recommended to compare against continuous latent variants and introduce soft quantization alternatives.

---

> ### Author Response · Authors · 2026-06-23
> **Response to Reviewer FwEB 1**
>
> > Thanks to Reviewer FwEB for fruitful comments. W, MA, and MI refer to Weakness, Major Concern, and Minor Concern, respectively. The corresponding revisions are marked in red in the revised manuscript. The RCIT and FG refer to the representational consistency of identical transformations and fixed grid, respectively
> ---
> ### W 1 & MA 1 & MI 2.
> > **(Response to MA 1)** We agree that RCIT is explicitly enforced rather than learned as an emergent property. As discussed in Sections 3.2 and 4, our goal is to examine the need for RCIT and identify suitable symmetry representations. We clarify this scope in the revised manuscript.
> > \\begin{array}{|c|c|c|c|c|c|c|c|}
> \\hline
> \\text{Method} & \\text{Supervision} & \\text{Architecture} & \\text{RCIT} & \\beta\\text{-VAE} & \\text{FVM} & \\text{MIG} & \\text{DCI} \\\\
> \\hline
> S^2/S\\text{-}\\beta\\text{-VAE} & \\checkmark & \\times & \\times & 75.71(\\pm11.16) & 70.34(\\pm14.01) & 7.59(\\pm5.80) & 11.52(\\pm6.14) \\\\
> \\text{CTFG-SP (w/o FG)} & \\checkmark & \\times & \\checkmark & 90.75(\\pm5.23) & 92.36(\\pm2.54) & 49.06(\\pm5.46) & 62.55(\\pm6.55)  \\\\
> \\text{CTFG-SP (w/o }\\mathcal{L}_c\\text{)} & \\checkmark & \\checkmark & \\triangle & 90.33(\\pm4.00) & 92.63(\\pm3.17) & 50.29(\\pm2.55) & 62.52(\\pm3.25) \\\\
> \\text{CTFG-SP (w/o }\\mathcal{L}_s\\text{)} & \\checkmark & \\checkmark & \\triangle & 67.75(\\pm7.67) & 58.02(\\pm6.85) & 3.02(\\pm0.86) & 6.47(\\pm1.50) \\\\
> \\text{CTFG-SP} & \\checkmark & \\checkmark & \\checkmark & \\textbf{91.40}(\\pm4.99) & \\textbf{93.74}(\\pm1.82) & \\textbf{51.02}(\\pm2.42) & \\textbf{64.69}(\\pm1.55)  \\\\
> \\hline
> \\end{array}
> >
> > $\textbf{Importance of Consistency Supervision}$ **(Response to Weakness 1 and Major 1)**: To isolate the effect of consistency supervision, we compare $S^2/S$-$\beta$-VAE [1] with CTFG-SP (w/o FG) under a controlled setting. Both methods use the same encoder, decoder, training data, factor labels, and amount of supervision. Neither method employs the fixed-grid component or any additional architectural inductive bias. CTFG-SP also receives no additional transformation annotations. Its transformation steps are computed solely from differences between the same factor labels used by $S^2/S$-$\beta$-VAE.
> >
> > **The key difference is how the same supervision is used**. $S^2/S$-$\beta$-VAE applies **image-level factor supervision** without explicitly enforcing the consistent representation of identical transformations. CTFG-SP (w/o FG) uses the factor-label differences to provide **transformation-level consistency supervision**. CTFG-SP (w/o FG) substantially improves all disentanglement metrics. This controlled comparison provides empirical evidence that explicitly **encouraging identical transformations to be represented consistently is an effective** inductive bias for learning disentangled representations.
> >
> > We added the quantitative comparison and latent-traversal visualization to the **Motivation paragraph in Section 3.2 of the revised manuscript**. They demonstrate that the lack of representational consistency produces entangled trajectories, whereas transformation-level consistency yields factor-specific changes. **Please see Section 3.2 for details**.
> >
> > $\\textbf{Performance Gains Persist Without Fixed-Grid Quantization}$ **(Response to W 1 and MI 2)**: To directly examine the reviewer’s concern, we evaluate **CTFG-SP (w/o FG) as a continuous** latent variant that retains consistency supervision. **Removing the fixed grid causes a small performance decrease** compared with the full model. CTFG-SP (w/o FG) also substantially outperforms the supervised $S^2/S$-$\beta$-VAE baseline across all disentanglement metrics. These results show that the fixed grid contributes to performance, **but does not alone account for the observed improvement**.
> >
> > $\\textbf{Relative Consistency Plays a Central Role}$ **(Response to W 1 and MA 1)**: Removing the canonical consistency loss $\mathcal{L}_c$ results in marginal degradation across the disentanglement metrics. This indicates that $\mathcal{L}_c$ provides auxiliary support and does not independently explain the improved performance.
> >
> > In contrast, removing the relative consistency loss $\mathcal{L}_s$ causes a much larger degradation. The $\beta$-VAE, FVM, MIG, and DCI scores decrease to $67.75$, $58.02$, $3.02$, and $6.47$, respectively. This result demonstrates that learning consistent relative transformations is closely related to the quality of the learned disentangled representation.
> >
> > **Overall**, the ablations suggest that the improvement cannot be explained solely by fixed-grid quantization or the canonical anchor loss, while **the large degradation without** $\mathcal{L}s$ **highlights the importance of relative consistency**. These results support representational consistency as a useful inductive bias in the considered settings. **We added these results to Section 6.3**.
> [1] Disentangling factors of variations using few labels, ICLR 2020.

---

> ### Author Response · Authors · 2026-06-23
> **Response to Reviewer FwEB 2**
>
> ### MA 1.
> > $\textbf{CTFG requires supervision to enforce RCIT}$: When CTFG (w/o FG) receives no supervision, it reduces to the standard $\beta$-VAE because no transformation-level signal is available to construct the RCIT objective. Thus, our current framework requires transformation-level supervision to enforce RCIT. As shown in Table 10 in the Appendix, reducing the known-label ratio generally degrades disentanglement performance. However, our scope is not to claim that RCIT emerges without supervision, but to investigate whether explicitly enforcing RCIT provides an effective inductive bias under supervised and weakly supervised settings. Learning RCIT in a fully unsupervised setting remains an important direction for future work.
>
> ### W 2 & MA 2.
>
> > $\textbf{Scope and Formal Conditions of the Analysis}$: We agree that the original presentation did not state the assumptions underlying our theoretical analysis with sufficient clarity. Our objective is not to establish a universal limitation for arbitrary symmetry groups or latent representations. Instead, we study suitable symmetry parameterizations for representing **finite cyclic factor symmetries in dimension-wise disentangled latent spaces**.
> >
> > These assumptions define our target problem. The finite cyclic assumption captures discrete factors with modulo structure in our controlled setting, while the dimension-wise assumption reflects our goal of representing each factor through a single latent dimension. Beyond the present analysis, allowing non-cyclic factors or multi-dimensional latent subspaces would be a possible extension direction in future work.
> >
> > To strengthen the formal justification, we have added Conditions 4.1--4.3 before the theoretical results in Section 4.1 of the revised paper. These conditions explicitly define the **1) finite cyclic factor symmetries, 2) dimension-wise latent actions, and 3) structure-preserving RCIT representations considered in our analysis**. We have also revised the theorem statements and proofs to explicitly reference these conditions and restricted our conclusions to this stated scope.
> >
> > $\textbf{Finite-Order Conditions and Strengthened Proofs}$: We refined Theorems 4.5 and 4.6 to focus on cyclic factors satisfying $|G^{F^i}|>2$. Multi-valued cyclic factors constitute a broad and practically relevant class of factor structures beyond the special binary case. The revised analysis therefore focuses on this broader class of non-binary cyclic factors.
> >
> > **To provide a complete formal justification**, we have added Lemma B.15. The lemma establishes that a finite-order matrix of the considered dimension-wise form is either the identity or an order-two transformation. Since an injective group homomorphism preserves the order of a generator, the lemma supports the revised results for factors satisfying $|G^{F^i}|>2$. These additions complete the revised proofs of Theorems 4.5 and 4.6.
> >
> > $\textbf{Reframing Corollary 4.5 from Symmetry-Structure Loss to Factor-State Information Loss}$ : We identified that the original Corollary 4.5 conflated the non-injectivity of the state-space mapping $b$ with the non-injectivity of the induced group homomorphism $\Gamma^\prime$. In particular, the original statement inferred that a surjective but non-injective $b$ necessarily yields $\Gamma^\prime(g_i)=\Gamma^\prime(g_j)$ for distinct symmetry elements. This implication does not hold in general.
> >
> > We therefore replaced the original corollary with Proposition 4.8 in the revised paper. The revised proposition makes **a state-level claim**. If two factor states differ in the $k$-th factor but are mapped to the same representation by $b\circ h_\phi$, then the resulting representation is not injective with respect to that factor and **cannot distinguish all of its values**. Thus, the revised result concerns the **loss of factor-state distinguishability**.
> >
> > We incorporated this revision into Proposition 4.8, updated the Type II limitation described in Section 5.1, and Appendix B.4  accordingly. These changes are marked as W 2 & MA 2 in the revised manuscript.
> ---
> ### W 3.
> > We agree that cyclic groups do not encompass all variations in real-world data. We acknowledge this restricted scope as a limitation of our work.
> >
> > Nevertheless, cyclic factors remain a meaningful and non-trivial setting. Our experiments show that existing approaches struggle to learn dimension-wise disentangled representations even on controlled benchmark datasets. This difficulty also affects disentanglement learning and compositional generalization.
> >
> > Our contribution is therefore not intended to provide a universal treatment of all possible variations. Instead, we investigate how factors of variation with cyclic structure can be represented appropriately and consistently. We clarify this scope and limitation in Section 8 of the revised manuscript.

---

> ### Author Response · Authors · 2026-06-23
> **Response to Reviewer FwEB 3**
>
> ### MI 1.
> > \\begin{array}{|c|c|c|c|c|c|}
> \\hline
> \\text{Loss type }(\\mathcal{L}_s) & \\text{Loss type }(\\mathcal{L}_c) & \\beta-\\text{VAE} & \\text{FVM} & \\text{MIG} & \\text{DCI} \\\\
> \\hline
> L1 & L1 & 90.00(\\pm2.58) & 92.91(\\pm 0.19) & \\textbf{51.92}(\\pm 2.40) & 57.81(\\pm 0.92) \\\\
> L1 & L2 & 90.67(\\pm2.31) & 90.75(\\pm 0.14) & 51.24(\\pm 2.12) & 57.29(\\pm 0.35) \\\\
> \\textbf{L2} & \\textbf{L1} & \\textbf{91.40}(\\pm 4.99) & \\textbf{93.74}(\\pm 1.82) & 51.02(\\pm 2.42) & \\textbf{64.69}(\\pm 1.55) \\\\
> L2 & L2 & 88.00(\\pm4.90) & 92.19(\\pm 3.70) & 40.92(\\pm 3.36) & 53.67(\\pm 1.58) \\\\
> \\hline
> \\end{array}
> >
> > $\textbf{Choice of L1 and L2 Losses}$: Equations 9 and 10 in the original manuscript were renumbered as Equations 6 and 7, respectively, in the revised manuscript. As shown in the ablation table, using L2 for $\mathcal{L}_s$ and L1 for $\mathcal{L}_c$ achieves the best overall performance among the evaluated combinations, obtaining the highest scores on three of the four metrics. We added this explanation and the ablation results to Section 6.3 of the revised manuscript (Table 7).

---

### Review · Reviewer_HQfe · 2026-05-11

**Summary Of Contributions:**

The paper argues that symmetry-based disentanglement should require not only equivariance, but also consistent latent representations of identical semantic transformations across different sample pairs. It proposes a bijective symmetry representation on the unit circle, together with a codebook-based latent vector space, and reports improved disentanglement and compositional generalization on benchmarks. The main strength is a clear and useful consistency criterion with broad empirical evaluation. The main weaknesses are a couple of somewhat unclear theoretical claims about representational consistency of identical transformations.

**Audience:**

Yes

**Audience Explanation:**

The paper will likely be of interest to researchers working on disentanglement learning and equivariance. The idea of representational consistency of identical transformations is natural and potentially useful, and the paper's controlled experiments suggest that this inductive bias can improve benchmark performance.

**Claims And Evidence:**

Yes

**Claims Explanation:**

Empirical results support the claim that the proposed framework yields improved disentanglement performance and strong compositional generalization.

Theorem 4.2 does not look correct. Definition 3.1 only requires that the same factor-space transformation $g^F$ map to the same latent transformation $\Gamma(g^F)$ independently of the source pair, and does not require the latent displacement $z_j-z_i$ to be constant.

- A nontrivial element of $GL'(2)$ gives a counterexample: encode $\mathbb Z_N$ states as $z_t=(\cos(2\pi t/N),\sin(2\pi t/N))$, and represent a k-step transformation by the rotation matrix $R_k$, where $z_{t+k}=R_k z_t$ for all t. Since
$R_k=\exp \left( [[0, -2\pi k/N], [2\pi k/N, 0] ] \right)$,
we have $R_k\in GL'(2)$, $R_k\neq I$ for nonzero k, and the same $R_k$ applies to every pair with the same transformation. Thus Theorem 4.2 seems to rule out representations that actually satisfy Definition 3.1, unless an additional restriction is imposed that consistency means constant additive displacement in a raw Euclidean coordinate.

Other theoretical claims seem sound to my best knowledge.

**Requested Changes:**

Could the authors clarify my question about the correctness of Theorem 4.2?

It would also be helpful for understanding Definition 3.1 if the authors could include an informal version or intuition of what representational consistency of identical transformations mean.

---

> ### Author Response · Authors · 2026-06-23
> **Response to Reviewer HQfe**
>
> > Thanks to Reviewer HQfe for fruitful comments. The **RICT** refers to “Representational Consistency of Identical Transformations”.
> ---
> ### Requested Changes
> > $\textbf{RCIT Requires Source-Independent Transformations, Not Constant Additive Displacements}$: RCIT does not generally require a constant additive displacement. To eliminate the ambiguity in the original presentation, **we revised Definition 3.1** to explicitly state that RCIT requires the same factor-space transformation to induce the same latent-space transformation independently of the source pair. **Please see Section 3.2 in the revised paper**.
> >
> > $\textbf{The Rotation Counterexample Falls Outside the Dimension-Wise Scope of Theorem 4.2. (Theorem 4.5 in the revised paper)}$: Theorem 4.5 concerns the **dimension-wise disentanglement** setting introduced in the paragraph “Lack of Analysis on Suitable Symmetry Parameterizations for Disentangled Representations” of Section 4. In this setting, each symmetry is required to **change only a single latent dimension**. Since the **rotation counterexample** acts on two or more latent dimensions, it **lies outside the scope of Theorem 4.5**. As this scope was not sufficiently clear, To avoid confusion and improve readability, we have added Conditions 1–3 to Section 4.1 of the revised paper.
> >
> > $\textbf{The Sparse Vector $v^\prime$ tests Dimension-Wise Realizability rather than Additive Displacement}$: We also clarify that the expression $e^{g}z=e\mathbf{I}gz+v^\prime$ in Proposition 4.1 (Proposition 4.4 in the revised version) was **not intended to define RCIT in terms of a constant additive displacement**. Instead, $v^\prime$ is introduced to examine **whether a nonzero sparse vector**, with only one nonzero dimension, **can be represented under the considered** $GL^\prime(n)$ and $GL(n)$ parameterizations while **satisfying the cyclic and dimension-wise conditions**. Propositions B.11 through Theorem B.19 in Appendix B.2 provide the detailed analysis showing that **no nonidentity element can realize this case**. We have clarified this point in the revised manuscript to prevent the equation from being interpreted as an additional requirement of Definition 3.1.

---

### Review · Reviewer_qe9b · 2026-06-09

**Summary Of Contributions:**

The paper studies symmetry-based disentangled representation learning, focusing on what it calls “representational consistency of identical transformations.” The concern they identify and aim to address is the following: two pairs of latent encodings may be related to each other via some group action of $g^F$ in factor space that generates the data. But in the latent space, these two pairs may be related to each other via different group actions $g^Z_1$ and $g^Z_2$. Representational consistency requires that an action of some group element $g^F$ is represented consistently in the latent space i.e. $g^Z_1=g^Z_2$ for all pairs of latents. They show that equivariance alone does not guarantee this and that widely used symmetry parameterizations don't satisfy it. They also develop an algorithm that induces equivariance while promoting representational consistency.

**Audience:**

Yes

**Audience Explanation:**

This work would be interesting to people working on disentangled representations and representation learning in general.

**Broader Impact Concerns:**

No, this work is mostly theoretical and does not have broader impact concerns.

**Claims And Evidence:**

Yes

**Claims Explanation:**

Yes, they describe the problem in definition 3.1. They prove that it exists within standard symmetry parameterizations in Section 4, propose a method to solve it in Section 5, and provide experimental validation in Section 6. Overall, I think it's an interesting paper.

However, some questions and concerns are not adequately addressed in the current draft of the paper.

## Section 3 (representational consistency)

1. I think my main concern with this paper and in this section was that the authors didn't sufficiently motivate why the lack of representational consistency is a problem in the first place. I would have liked to have seen a more detailed discussion of this beyond the appeals to intuition, like in the introduction. Ideally, this would either be a clear theoretical motivation where we lack consistency, provably causes some kind of failure, or empirical evidence, even in toy settings, that shows that the lack of consistency leads to some kind of clear visible failure in learning disentangled representations.
2. I think definition 3.1 is not written clearly. I could understand what the authors mean by looking at Figure 1, but the clarity and rigor of definition 3.1 should be improved. It is written almost verbally rather than a clear mathematical definition, such as the ones in section 2.
3. Relatedly, there are a few places I noticed where the notation is not consistent/overloaded. For example, the notation for the number of factors switches between $n$, $k$, and $|F^i|$, $\alpha$ is overloaded, and $k$ is overloaded.

## Section 6 (Experiments)
1. Table 1 and Table 2 show better disentanglement performance across various metrics. But why should enforcing representational consistency lead to better disentanglement? This was unclear. As a result, it is unclear to me whether the improved disentanglement is due to consistency, fixed grid, Cayley transform, or supervision. Some kind of ablation is required to verify weather consistency and disentanglement performance are related. For example, the authors could remove $L_s$ and $L_c$ or deliberately introduce pair-dependent inconsistent transformation codes for identical semantic changes.

**Requested Changes:**

## Major
1. Mainly, I would like a better motivation for why one should care about representational consistency. As stated above this could be either theoretical evidence or a small experiment in a toy setting that clearly demonstrates the issues caused by the lack of representational consistency.
2. The experiments should do a better job of validating whether the improvements in disentanglement perofrmance is actually due to representational consistency or if it is caused by something else.

## Minor
1. Improve definition 3.1 to make it clearer and more rigorous.
2. Improve notational consistency and don't use the same notation for different things.

---

> ### Author Response · Authors · 2026-06-23
> **Response to Reviewer qe9b 1**
>
> > Thanks to Reviewer qe9b for fruitful comments. S, MA, and MI refer to Section, Major requested changes, and Minor requested changes, respectively. The corresponding revisions are marked in blue in the revised manuscript.
> ---
> ### S 3-1 & MA 1.
> > We clarified the problem and its motivation in the Introduction section and added empirical evidence in Section 3.2 of the revised paper to illustrate how representational inconsistency can lead to a clear failure in learning disentangled representations. The RCIT and FG refer to the representational consistency of identical transformations and fixed grid, respectively.
> > \\begin{array}{|c|c|c|c|c|c|c|c|}
> \\hline
> \\text{Method} & \\text{Supervision} & \\text{Architecture} & \\text{RCIT} & \\beta\\text{-VAE} & \\text{FVM} & \\text{MIG} & \\text{DCI} \\\\
> \\hline
> S^2/S\\text{-}\\beta\\text{-VAE} & \\checkmark & \\times & \\times & 75.71(\\pm11.16) & 70.34(\\pm14.01) & 7.59(\\pm5.80) & 11.52(\\pm6.14) \\\\
> \\text{CTFG-SP (w/o FG)} & \\checkmark & \\times & \\checkmark & \\textbf{90.75}(\\pm5.23) & \\textbf{92.36}(\\pm2.54) & \\textbf{49.06}(\\pm5.46) & \\textbf{62.55}(\\pm6.55) \\\\
> \\hline
> \\end{array}
> >
> > $\textbf{Importance of Consistency Supervision}$: To isolate the effect of consistency supervision, we compare $S^2/S$-$\beta$-VAE [1] with CTFG-SP (w/o FG) under a controlled setting. Both methods use the same encoder, decoder, training data, factor labels, and amount of supervision. Neither method employs the fixed-grid component or any additional architectural inductive bias. CTFG-SP also receives no additional transformation annotations. Its transformation steps are computed solely from differences between the same factor labels used by $S^2/S$-$\beta$-VAE.
> >
> > **The key difference is how the same supervision is used**. $S^2/S$-$\beta$-VAE applies **image-level factor supervision** without explicitly enforcing the consistent representation of identical transformations. CTFG-SP (w/o FG) uses the factor-label differences to provide **transformation-level consistency supervision**. CTFG-SP (w/o FG) substantially improves all disentanglement metrics. This controlled comparison provides empirical evidence that explicitly **encouraging identical transformations to be represented consistently is an effective** inductive bias for learning disentangled representations.
> > We added the quantitative comparison and latent-traversal visualization to the **Motivation paragraph in Section 3.2 of the revised manuscript**. They demonstrate that the lack of representational consistency produces entangled trajectories, whereas transformation-level consistency yields factor-specific changes. **Please see Section 3.2 for details**.
> >
> > [1] Disentangling factors of variations using few labels, ICLR 2020.

---

> ### Author Response · Authors · 2026-06-23
> **Response to Reviewer qe9b 2**
>
> ### S 3-2 & MI 1
> > We revised the definition 3.1 clearly, and we add it at Section 3.2 in the revised version paper. We highlighted the modified part of Figure 1, Section 2, and Section 3.2. in the revised paper.
> We agree that the previous definition did not state the consistency condition explicitly. Previously, RCIT was described through the following two equivariance relations:
> $$
> q_\phi \circ \Omega(g^F \cdot fv_i)
> =
> \Gamma(g^F) \cdot q_\phi \circ \Omega(fv_i),
> $$
> $$
> q_\phi \circ \Omega(g^F \cdot fv_n)
> =
> \Gamma(g^F) \cdot q_\phi \circ \Omega(fv_n).
> $$
> Because the same $\Gamma(g^F)$ was used for both pairs, representational consistency was implicitly built into the notation rather than stated as an explicit condition. In the revised Definition3.1, we separately define the pair-induced latent transformations $g^Z_{ij}$ and $g^Z_{nm}$ through
> $$
> z_j = g^Z_{ij} \cdot z_i,
> \qquad
> z_m = g^Z_{nm} \cdot z_n,
> $$
> and explicitly define RCIT as
> $$
> g^Z_{ij} = g^Z_{nm}.
> $$
> This reformulation makes clear which latent transformations are compared and under what condition representational consistency holds. The corresponding revisions in Figure 1 and Sections 2 and 3.2 are highlighted. Here is the revised Definition 3.1:
> >
> > - Let $G^F$ be a group acting on the factor space $\mathcal{F}$, and let $G^Z$ be a group acting on the latent space $\mathcal{Z}$.
> >
> > - $\textbf{Representational consistency of identical transformations}$:
> Let $g^F\in G^F$ be a factor space transformation, and consider two pairs
> $(fv_i,fv_j),(fv_n,fv_m)\in\mathcal{F}\times\mathcal{F}$ generated by the same transformation $g^F$:
> $ fv_j = g^F\cdot fv_i$ and $fv_m = g^F\cdot fv_n.$
> Let $z_i,z_j,z_n,z_m\in\mathcal{Z}$ be their latent representations:
> $ z_i=h^\prime_\phi(fv_i), z_j=h^\prime_\phi(fv_j), z_n=h^\prime_\phi(fv_n),$
> and $z_m=h^\prime_\phi(fv_m)$, where $h^\prime_\phi: \mathcal{F} \rightarrow \mathcal{Z}$.
> Let $g^Z_{ij}, \, g^Z_{nm}\in G^Z$ denote the pair-induced latent transformations satisfying $z_j = g^Z_{ij}\cdot z_i$ and $z_m = g^Z_{nm}\cdot z_n.$
> We say that the **same factor space transformation $g^F$** is represented consistently across two pairs **if the corresponding pair-induced latent transformations are identical: $g^Z_{ij}=g^Z_{nm}$**.
> We refer to this property as the **representational consistency of identical transformations** (RCIT).
> Accordingly, the representation map $h^\prime_\phi$ is said to satisfy RCIT if $g^Z_{ij}=g^Z_{nm}$ holds for every $g^F\in G^F$ and for all pairs in $\mathcal{F}\times\mathcal{F}$ generated by the same $g^F$.
> ---
> ### S 3-3 & MI 2.
> > We made the following notational revisions and marked them in blue in the revised paper **(Section 4 and Section 5)**:
> >
> > Group action: $\alpha(\cdot,\cdot) \rightarrow act(\cdot,\cdot)$, to avoid confusion with the hyperparameter $\alpha$.
> >
> > Number of factors: $i \rightarrow k$, to distinguish it from $i$, which denotes the $i$-th dimension.
> >
> > Rotation index for the cyclic group: $\frac{2\pi}{N}k \rightarrow \frac{2\pi}{N}k^\prime$, to distinguish it from the number of factors $k$.
> >
> > $\theta_i^k, z_i^k \rightarrow \theta^i, z^i$, for simplicity.

---

> ### Author Response · Authors · 2026-06-23
> **Response to Reviewer qe9b 3**
>
> ### S 6 & MA 2.
> > \\begin{array}{|c|c|c|c|c|c|c|c|c|c|}
> \\hline
> \\text{Method} & \\text{Supervision} & \\text{Architecture} & \\text{RCIT} & \\beta\\text{-VAE} & \\text{FVM} & \\text{MIG} & \\text{DCI} & \\text{R2E} & \\text{R2R} \\\\
> \\hline
> \\text{CTFG-SP (w/o FG)} & \\checkmark & \\times & \\checkmark & 90.75(\\pm5.23) & 92.36(\\pm2.54) & 49.06(\\pm5.46) & 62.55(\\pm6.55) & 8.87(\\pm0.91) & 138.80(\\pm17.19)  \\\\
> \\text{CTFG-SP (w/o }\\mathcal{L}_c\\text{)} & \\checkmark & \\checkmark & \\triangle & 90.33(\\pm4.00) & 92.63(\\pm3.17) & 50.29(\\pm2.55) & 62.52(\\pm3.25) & 8.71(\\pm0.97) & 151.30(\\pm19.09) \\\\
> \\text{CTFG-SP (w/o }\\mathcal{L}_s\\text{)} & \\checkmark & \\checkmark & \\triangle & 67.75(\\pm7.67) & 58.02(\\pm6.85) & 3.02(\\pm0.86) & 6.47(\\pm1.50) & 9.02(\\pm1.12)& 159.24(\\pm23.51) \\\\
> \\text{CTFG-SP} & \\checkmark & \\checkmark & \\checkmark & \\textbf{91.40}(\\pm4.99) & \\textbf{93.74}(\\pm1.82) & \\textbf{51.02}(\\pm2.42) & \\textbf{64.69}(\\pm1.55) & \\textbf{7.24}(\\pm0.94) & \\textbf{135.70}(\\pm16.48) \\\\
> \\hline
> \\end{array}
> >
> > $\\textbf{Importance of Representational Consistency of Identical Transformation}$: The **relatively small performance drop** after **removing the fixed grid or the absolute consistency loss $\mathcal{L}_c$** further suggests that these components mainly provide auxiliary support for consistency, rather than being the sole source of the observed consistency.
> Specifically, removing the fixed grid only slightly decreases the performance compared to the CTFG-SP model, while CTFG-SP (w/o FG) still substantially outperforms the supervised $S^2/S$-$\beta$-VAE baseline.
> Similarly, removing $\mathcal{L}_c$ results in only marginal degradation across the disentanglement metrics.
> These observations indicate that **neither the fixed grid nor $\mathcal{L}_c$ alone explains the improved disentanglement performance**.
> >
> > In contrast, **removing the main consistency-related component causes a much larger degradation**, reducing the beta-VAE, FVM, MIG, and DCI scores to $67.56$, $58.02$, $3.02$, and $6.47$, respectively.
> This sharp performance drop indicates that representational consistency is closely related to the quality of the learned disentangled representation.
> Overall, the ablation results provide evidence that representational consistency is not merely an artifact of fixed-grid quantization or the absolute consistency loss, but a key inductive bias that contributes to both disentanglement learning and compositional generalization.
> >
> > We add these results **in Section 6.3. Discussion in the revised paper**.
> ---
> > $\textbf{Empirical Motivation for RCIT in Compositional Generalization}$:
> An identical semantic transformation in factor space acts as a reusable rule independently of the source state. RCIT transfers this property to the latent space, allowing the same latent action to be reused for unseen compositions. Under a controlled comparison using the same factor-label information without the fixed grid, CTFG-SP (w/o FG) reduces the R2E and R2R errors of $S^2/S$-$\beta$-VAE from $10.54$ and $164.29$ to $8.87$ and $138.80$, respectively. The full CTFG-SP further reduces them to $7.24$ and $135.70$, supporting RCIT as a useful inductive bias for compositional generalization. We added in Section 6.3 of the revised paper.